# Sinking microplastics in the water column: simulations in the Mediterranean Sea

Rebeca de la Fuente[1], Gábor Drótos[1,2], Emilio Hernández-García[1], Cristóbal López[1], and Erik van Sebille[3,4]

[1]IFISC (CSIC-UIB), Campus Universitat de les Illes Balears, Palma de Mallorca, Spain
[2]MTA–ELTE Theoretical Physics Research Group, Budapest, Hungary
[3]Institute for Marine and Atmospheric Research, Utrecht University, Utrecht, the Netherlands
[4]Centre for Complex Systems Studies, Utrecht University, Utrecht, the Netherlands

**Correspondence:** Emilio Hernández-García (emilio@ifisc.uib-csic.es)

**Abstract.** We study the vertical dispersion and distribution of negatively buoyant rigid microplastics within a realistic circulation model of the Mediterranean sea. We first propose an equation describing their idealized dynamics. In that framework, we evaluate the importance of some relevant physical effects: inertia, Coriolis force, small-scale turbulence and variable seawater density, and bound the relative error of simplifying the dynamics to a constant sinking velocity added to a large-scale velocity field. We then calculate the amount and vertical distribution of microplastic particles on the water column of the open ocean if their release from the sea surface is continuous at rates compatible with observations in the Mediterranean. The vertical distribution is found to be almost uniform with depth for the majority of our parameter range. Transient distributions from flash releases reveal a non-Gaussian character of the dispersion and various diffusion laws, both normal and anomalous. The origin of these behaviors is explored in terms of horizontal and vertical flow organization.

## 1 Introduction

Approximately 8 million tonnes of plastics end up in the oceans every year (Jambeck et al., 2015). Nevertheless, only a very small percentage, around 1%, remains on the surface (van Sebille et al., 2015; Choy et al., 2019). The rest leaves the surface of the ocean (Ballent et al., 2013; van Sebille et al., 2020) through beaching (Turner and Holmes, 2011), biofouling (Ye and Andrady, 1991; Chubarenko et al., 2016; Kooi et al., 2017) or sinking (Erni-Cassola et al., 2019), but also wind-driven mixing presumably leads to an underestimation for the amount of particles remaining close to sea surface (Kukulka et al., 2012; Enders et al., 2015; Suaria et al., 2016; Poulain et al., 2018). The distribution of plastic pollution in the sea is poorly understood at present but would be crucial to properly evaluate the exposure of marine biota to this material, and formulate strategies for cleaning the oceans (Horton and Dixon, 2018).

Floating plastics and those that have beached or sedimented on the seafloor are relatively well studied through field campaigns (although explanation is missing for many findings; Andrady, 2017; Erni-Cassola et al., 2019; Kane and Clare, 2019). In contrast, the presence of plastics within the water column has received less attention, and many surveys in this realm are restricted to so-called underway samples, a few meters below the surface (e.g., Enders et al., 2015). However, e.g., Choy et al. (2019) reported that below the mixed layer and down to 1000 m depth in Monterey Bay, concentrations of plastics are larger than at the surface (Thompson et al., 2004; Hidalgo-Ruz et al., 2012). Egger et al. (2020) found more plastic between 5 m and 2000 m below the North Pacific Garbage Patch than at the surface. These findings turn out to mostly concern plastic pieces that, according to their nominal material density, would be classified as positively buoyant (Egger et al., 2020).

In this paper, we focus on a certain class of plastic particles, negatively buoyant rigid microplastics, excluding very small size, and we estimate their vertical distribution through the water column and their amount in the Mediterranean Sea. Microplastic particles are among the most important contributors to marine plastic pollution (Arthur et al., 2009). Closely following the work of Monroy et al. (2017) for sinking biogenic particles but choosing particle properties to correspond to those of negatively buoyant microplastics, we first justify the use of a simplified equation of motion, in which the plastic particle velocity is the sum of the ambient flow velocity and a sinking velocity depending on particle and water characteristics. In particular, we estimate the impact of some corrections to this simple dynamics and evaluate in detail the influence of the spatial variation of the seawater density on the plastic dispersion and sinking characteristics. For our Mediterranean case study, the impact of the varying seawater density on particle trajectories can be comparable to the estimated effect of the neglected small scales below the hydrodynamical model's resolution.

We then estimate the amount of microplastic particles in the water column of the open Mediterranean. Our estimates rely on a uniform vertical distribution, which is confirmed by our numerical simulations to be a good approximation for fast-sinking particles. This can be explained by a simple model in which released particles sink with a constant velocity. Detailed consideration of the transient dynamics identifies small non-Gaussian vertical dispersion around this simple sinking behavior, with transitions between anomalous and normal effective diffusion.

## 2  Types of microplastics in the water column

The dynamics and the fate of microplastics in the ocean are largely determined by their material density (Erni-Cassola et al., 2019). However, shape, size and rigidity are also relevant properties, characteristic transport pathways to the water column being different for different particle types.

Typically, positively buoyant plastic types will remain floating at the sea surface or close to it, and then will not contribute to the microplastic content in the water column, the topic we are interested in this paper. However, it has been documented experimentally that biofouling may increase sinking rates of particles up to 81% and enhances sedimentation (Kaiser et al., 2017). So, a class of high abundance and mass may be represented by nearly neutrally buoyant microplastic particles that are generated by biofouling (Ye and Andrady, 1991; Chubarenko et al., 2016) from positively buoyant plastic types or by other mechanisms of aggregation with organic matter, especially for small particle sizes (Kooi et al., 2017).

In fact, the fallout from the North Pacific Garbage Patch almost entirely consists of plastic types nominally less dense than water (Egger et al., 2020). Although some of these immersed particles finally reach the sea bottom, their proportion in sedimented plastic is minor except for the immediate vicinity of coasts where water is shallow. Most of these particles remain in the photic zone (Mountford and Morales Maqueda, 2019; Wichmann et al., 2019; Soto-Navarro et al., 2020). This suggests that reverse processes could also take place after biofouling and that the dynamics of such particles is complicated (Kooi et al., 2017; Erni-Cassola et al., 2019).

Particles denser than seawater dominantly accumulate at the sea bottom (Mountford and Morales Maqueda, 2019). A mechanism by which microplastics denser than water can also be present within the water column is the finite time taken by them to reach the bottom. Under continuous release at the surface and sedimentation at the bottom, the transient falling would lead to a steady distribution for the amount of plastic in the water column at any given time, and this distribution has never been estimated. Note that the Eulerian methodology of Mountford and Morales Maqueda (2019), treating sedimentation (i.e., deposition on the seafloor) by parametrization and thus leaving particles in the water column indefinitely long, is not suitable for this estimation. One aim of this paper is to explore this distribution by means of Lagrangian simulations.

There are different classes of microplastic particles denser than seawater. For example, dense synthetic microfibers have been found to strongly dominate in sediment samples far from the coast (Woodall et al., 2014; Fischer et al., 2015; Bergmann et al., 2017; Martin et al., 2017; Peng et al., 2018), and have been detected in large proportions in deep-water samples and sediment traps in the open sea as well (Bagaev et al., 2017; Kanhai et al., 2018; Peng et al., 2018; Reineccius et al., 2020). Mostly originating from land-based sources (Dris et al., 2016; Carr, 2017; Gago et al., 2018; Wright et al., 2020), it is not obvious to explain their abundance on abyssal oceanic plains (Kane and Clare, 2019). Maritime-activity sources (Gago et al., 2018) can contribute to that. Another reason could be that their special and deformable shape results in a strongly reduced settling velocity (Bagaev et al., 2017) that allows long distance horizontal transport (Nooteboom et al., 2020). In any case, it is difficult to estimate the amount of microfibers in the oceans due to sampling issues and to their absence from statistics of mismanaged plastic waste (Carr, 2017; Barrows et al., 2018), and we will not consider them further in this paper. We also disregard films, which are only sporadically encountered in the open ocean (Bagaev et al., 2017) and thus have moderate importance.

We concentrate in the following on dense rigid microplastic particles. The most abundant particles of this class are fragments (e.g., Martin et al., 2017; Peng et al., 2018), which have an irregular shape, but their extension is usually comparable in the three dimensions. Experimental estimates for the settling velocities of irregular fragments or other nonspherical particles have suggested considerable deviations from values predicted by the Stokes law (Kowalski et al., 2016; Khatmullina and Isachenko, 2017; Kaiser et al., 2019), so that it is unclear how a precise full equation of motion should be constructed. For a qualitative exploration of particle transport through the water column, we will argue in Sect. 4.1 and App. A that the Maxey–Riley–Gatignol (MRG) equation (Maxey and Riley, 1983) should be appropriate for a reasonably wide range of such particles.

Whatever their precise equation of motion is, these sinking particles (directly detected by Bagaev et al. (2017) and Peng et al. (2018)) are thought to reach the seafloor relatively fast (Chubarenko et al., 2016; Kane and Clare, 2019; Soto-Navarro et al., 2020), landing within horizontal distances of tens of kilometers from their surface location of release (see Sect. 4.2 and App. B). One consequence of their fast sinking is the absence of almost any fragmentation after they leave the sea surface (Andrady,

2015; Corcoran, 2015), and the influence of biological processes on the particles' properties should also be moderate, leaving their size and shape intact during sinking. Note that, in contrast to the case of floating plastics (Kooi et al., 2017; Kvale et al., 2020), interaction of sinking plastics with particulate matter of biological origin appears to be moderate. This is according to the absence of a need to disassemble microplastic pieces from biological aggregates during sample processing as described by Bagaev et al. (2017). Note, however, that experimental results by Michels et al. (2018) indicate that aggregation with organic material might occur within a sufficiently short time at surface layers, which would likely lead to increased sinking velocities (Long et al., 2015). Transport by bottom currents (Kane and Clare, 2019; Kane et al., 2020) is important for explaining their distribution in sediments after coastal release. However, the statements above imply that the dense rigid microplastic content of samples from deep-sea trenches, abyssal plains (van Cauwenberghe et al., 2013; Fischer et al., 2015; Peng et al., 2018; Kane and Clare, 2019) must originate from sources at the surface of the open sea rather than from coastal inputs.

While methodological issues make the quantification of abundance difficult (Song et al., 2014; Andrady, 2015; Filella, 2015; Lindeque et al., 2020), negatively buoyant microplastic fragments have indeed been found in surface and near-surface samples of the open waters of the Mediterranean Sea (Suaria et al., 2016) and the Atlantic Ocean (Enders et al., 2015), respectively, from which they can contribute to microplastic content of the water column and deep-sea sediments (Fischer et al., 2015; Bagaev et al., 2017). Horizontal transport of these particles can be carried out by marine organisms, and spontaneous attachment to pieces of positive buoyancy is a further possibility but is not yet discussed in the literature. Composite pieces of debris or those that contain trapped air (including foams in some cases) may also represent a source of microplastic ending at the water column (Andrady, 2015). However, most of such particles are presumably released by local maritime activity. An example of this are flakes of paint and structural material from boats and ships, which contain negatively buoyant alkyds and poly-(acrylate/styrene). Despite the particle's high density, large amounts of them may be found in the sea surface microlayer where surface tension keeps them floating (Song et al., 2014). The range of horizontal transport of these particles at the sea surface is unclear, but expected to be restricted to short distances because sinking from the sea surface microlayer is considerable, especially in waters disturbed by waves (Hardy, 1982; Stolle et al., 2010).

While the idea of Kooi and Koelmans (2019) to treat all plastic particles together by means of continuous distributions is appealing, the above considerations strongly favor the separate treatment of positively buoyant pieces, negatively buoyant microfibers, and negatively buoyant rigid particles of sufficiently big size, since these classes have very different dynamics and sources. In the following we concentrate on the properties, amount and dynamics of particles of the last class.

## 3 Considerations for modeling negatively buoyant rigid microplastics

### 3.1 Physical properties

From a meta-analysis of 39 previous studies, Erni-Cassola et al. (2019) established the proportion of the most abundant polymer types discharged into water bodies: PE (polyethylene, 23 %), PP (polypropylene, 13 %), PS (polystyrene, 4 %) and PP&A (group of polymer types formed by polyesters, PEST, polyamide, PA and acrylics, 13 %). Note that these proportions do not distinguish between different regions (e.g., coastal region or open water; even inland water bodies of urban environments are

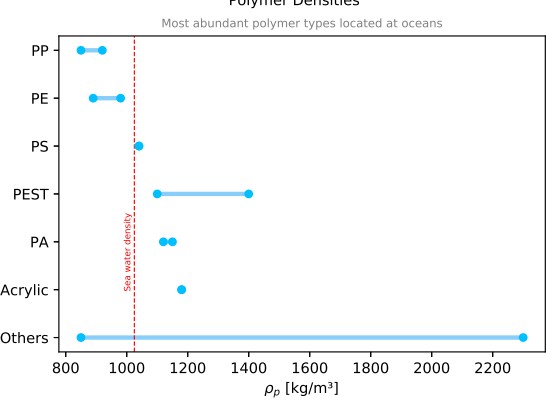

**Figure 1.** Polymer densities for the most abundant microplastics identified in water bodies (Erni-Cassola et al., 2019).

included in the analysis) and between the particle types (size range and shape) concerned in the different studies. We organize these polymer types according to their density (Chubarenko et al., 2016; Andrady, 2017; Erni-Cassola et al., 2019) in Fig. 1: PP between $850 - 920 \; kg/m^3$, PE $890 - 980 \; kg/m^3$, PS with $1040 \; kg/m^3$ (excluding its foamed version), PEST in the range $1100 - 1400 \; kg/m^3$, PA within $1120 - 1150 \; kg/m^3$ and acrylic with $1180 \; kg/m^3$. There is also some less abundant plastic like polytetrafluoroethylene (PTFE) which has higher densities, in the range $2100 - 2300 \; kg/m^3$.

Thus, the full range of microplastic particle densities in the ocean, denoted here as $\rho_p$, is $850 - 2300 \; kg/m^3$, and most of them have densities within the interval $850 - 1400 \; kg/m^3$. This has to be compared with the seawater density, which close to the surface has a conventional mean value of $\rho_f = 1025 \; km/m^3$ (red line in Fig. 1) and changes around $1\%$ from the surface to the sea bottom. Since we are interested in sinking material, and for the sake of maximal practicality, we restrict our study to microplastics of densities $1025 \; kg/m^3 < \rho_f < 1400 \; kg/m^3$.

Another relevant property of plastic particles is their size. By a widely accepted definition, microplastics are particles with a diameter less than $5 \; mm$ without any lower limit (Arthur et al., 2009). Some observations at the ocean surface show that the most common diameter is around $1 \; mm$ (Cózar et al., 2014, 2015), with an exponential decay with increasing diameter up to $100 \; mm$. However, the absence of this peak in other studies that show an increasing abundance with further decreasing size (Enders et al., 2015; Suaria et al., 2016; Erni-Cassola et al., 2017) suggests (Song et al., 2014; Andrady, 2015; Erni-Cassola et al., 2017; Bond et al., 2018; Lindeque et al., 2020) the need for new technologies in sampling methods (which usually use trawl nets with a mesh size around $0.3 \; mm$) and especially for the adaptation of careful and standardized analysis procedures to avoid artifacts (Filella, 2015).

Field data about distributions of size and quantifiers of shape for negatively buoyant rigid particles in the water column or deep-sea sediments are not available to date to the best of our knowledge, except in the Artic for Bergmann et al. (2017). However, their results may not apply to the majority of the oceans because of the very special dynamics provided by melting and freezing of sea ice (Bergmann et al., 2017). Data from Bergmann et al. (2017) and Song et al. (2014) about unspecified sedimented fragments and paint particles, respectively, exhibit an increasing abundance with decreasing size, most particles

being smaller than $0.05\ mm$. Laboratory findings about surface degradation of individual particles also indicate such a tendency (Song et al., 2017). Thus, these findings seem to indicate the prominent presence of small pieces of plastic. Nevertheless the observations of Bagaev et al (2017), Kanhai et al. (2018) and Peng et al. (2018) do not indicate this abundance of small particles.

For these reasons, we will disregard particles of extremely small size. To keep our qualitative study sufficiently simple, we will consider all our modelled particles to have a radius $a = 0.05\ mm$ (a diameter of $0.1\ mm$). This is a rather small size, but still within the commonly measured ranges. As we will discuss in Section 4.1, this radius is well within the validity range of the MRG equation.

## 3.2 Source estimation

In this subsection we indicate the total amount of dense microplastics entering the water column in open waters of the Mediterranean. Despite the correlation of plastic source with coastal population density, the rapid fragmentation of small particles along the shoreline (Pedrotti et al., 2016) and the seasonal variability of spatial distribution of floating particles (Macias et al., 2019), we focus on local maritime activity and exclude direct release from surface accumulation areas or the coast, either from urban areas or from rivers. The estimations are based on the results of Kaandorp et al. (2020). They provide a total amount of yearly plastic release into the Mediterranean in the range $2200 - 4000$ tonnes, from which around 37% corresponds to negatively buoyant plastic, and 6% are due to maritime activity. This 37% agrees well with previous global estimations (Lebreton et al., 2018).

We will take these numbers, 4000 tonnes per year, 37% of sinking particles, and the proportion of direct release by maritime activity (6%) to obtain in Sect. 4.3 an estimate for the basin-wide yearly release of negatively buoyant sphere-like microplastics in the open Mediterranean. Note that we choose the upper bound, 4000 tonnes per year, in order to account for the considerable amount of unregistered particles.

## 3.3 Dynamics

A standard modeling approach (Siegel and Deuser, 1997; Monroy et al., 2017; Liu et al., 2018; Monroy et al., 2019) for the transport of noninteracting sinking particles is to consider the time-dependent particle velocity $v$ as the combination of the ambient fluid flow $u$ and a settling velocity $v_s$ as:

$$v = u + v_s \,, \tag{1}$$

with

$$v_s = (1 - \beta)g\tau_p \,, \ \beta = \frac{3\rho_f}{2\rho_p + \rho_f} \,, \text{ and } \tau_p = \frac{a^2}{3\beta\nu} \,. \tag{2}$$

$g$ denotes the gravitational acceleration vector, pointing downwards; $\beta$ is a parameter depending on the particle and the fluid densities, $\rho_p$ and $\rho_f$, respectively. Particles heavier than water have $\beta < 1$, and $\beta = 1$ for neutrally buoyant particles. The expression given for $\beta$ assumes spherical particles. $\tau_p$ is the Stokes time, i.e., the characteristic response time of the particle

to changes in the flow, where $a$ is the radius of the particle and $\nu$ the kinematic viscosity of the fluid. Although Eq. (1) is commonly used, we are not aware of a systematic justification of it in the microplastics context. This will be done in Section 4.1.

## 3.4   Numerical procedures

For the flow velocity $\boldsymbol{u}$ we use a 3D velocity field from NEMO (Nucleus for European Modelling of the Ocean), which
implements a horizontal resolution of 1/12 degrees and 75 s-levels in the vertical with updates data every 5 days (Madec, 2008; Madec and Imbard, 1996). Salinity and temperature are also extracted from that model. The Parcels Lagrangian framework (Delandmeter and van Sebille, 2019) is used to integrate the particle trajectories from Eq. (1) or more complex ones to be considered in Sect. 4.1. Typical numerical experiments to obtain the results presented below consist of distributing a large number $N$ of particles in a horizontal layer over the whole Mediterranean on the nodes of a sinusoidal-projection grid (Seong
et al., 2002), so that their release is with uniform horizontal density. We locate this input source at 1 m depth to avoid surface boundary conditions. After particles are released at some initial date, in a so-called flash release, they evolve under equations of motion such as Eq. (1), and the statistics of the resulting particle cloud are analyzed.

## 4   Results

### 4.1   Range of validity of Eq. (1)

We next show, closely following the treatment of Monroy et al. (2017) for biogenic particles, that possible inertial effects that would correct Eq. (1) are negligible for the sizes and densities of typical dense microplastics. To this end, similarly to many other studies (Michaelides, 2003; Balkovsky et al., 2001; Cartwright et al., 2010; Haller and Sapsis, 2008), we start by choosing the simplified standard form of the more fundamental Maxey–Riley–Gatignol (MRG) equation (Maxey and Riley, 1983), and analyze under which conditions it is valid for microplastic transport. After finding the MRG equation to be valid
for an important range of microplastic particles, we will explore its relationship with Eq. (1).

The simplified MRG equation gives the velocity $\boldsymbol{v}(t)$ of a very small spherical particle in the presence of an external flow $\boldsymbol{u}(t)$ as

$$\frac{d\boldsymbol{v}}{dt} = \beta \frac{D\boldsymbol{u}}{Dt} + \frac{\boldsymbol{u} - \boldsymbol{v} + \boldsymbol{v}_s}{\tau_p}. \tag{3}$$

Beyond sphericity, two conditions are needed for the validity of Eq. (3) (Monroy et al., 2017; Maxey and Riley, 1983): a)
the particle radius, $a$, has to be much smaller than the Kolmogorov length scale $\eta$ of the flow, which has values in the range $0.3\,mm < \eta < 2\,mm$ for wind-driven turbulence in the upper ocean (Jiménez, 1997); b) the particle Reynolds number $Re_p = \frac{a|\boldsymbol{v}-\boldsymbol{u}|}{\nu} \approx \frac{av_s}{\nu}$ should satisfy $Re_p \ll 1$. Note that this last condition imposes restrictions on the values of the particles' density and size, partially via the settling velocity $v_s = |\boldsymbol{v}_s|$. For the most abundant sinking microplastics, i.e., with densities $\rho_p = 1025 - 1400\,kg/m^3$, we now determine the range of validity of Eq. (3) assuming $\nu = 1.15 \times 10^{-6}\,m^2/s$ and $\rho_f = 1025\,kg/m^3$

to be fixed. This gives $\beta$ in the range $0.8 - 1$. The possibility of small changes in the seawater density as the particle sinks, which translates to variations in $\boldsymbol{v}_s$, will also be analyzed in Section 4.2.

In Fig. 2 we show a diagram with the settling velocities and particle sizes for which Eq. (3) is valid. We plot the minimal value of the Kolmogorov scale $\eta = 0.3 \ mm$ with the red line (Jiménez, 1997), and $Re_p = 1$ with a black line, which bound the area of validity (shaded in the plot). We also indicate $v_s$ as a function of $a$ for $\beta = 0.8$ with the blue curve, corresponding to the upper bound to $v_s$ for typical microplastic densities. In total, the zone with *soft* shading in Fig. 2 represents a parameter region where Eq. (3) applies for particles with $\beta < 0.8$ (i.e., particles falling faster than the typical ones), whereas the area of our interest, corresponding to $\beta \geq 0.8$, is represented by a *dark* shading, denoting the typical plastic sizes and corresponding settling velocities for which the equation is valid. As a rule of thumb, in a typical situation, validity of Eq. (3) requires $v_s < 0.01 \ m/s$ and $a < 0.3 \ mm$. As discussed in Section 3.1, information about particles in the validity range is particularly sparse for surface waters because of the usual sampling techniques, but sediment data indicates the prevalence of sufficiently small particles. Furthermore, in sufficiently calm waters, the Kolmogorov scale is larger (of the order of millimeters, Jiménez (1997)), so that $a$ can be increased to this size without compromising the equation validity. These estimates of the Kolmogorov scale anyway assume wind-driven turbulence and are thus restricted to the mixed layer (Jiménez, 1997), below which $\eta$ is undoubtedly larger. Deviations from a spherical shape may lead to a more complicated motion than that described by the MRG equation, especially through particle rotation (Voth and Soldati, 2017). In App. A, we present quantitative arguments for the applicability of the MRG equation to rigid microplastic particles of common shapes in the parameter ranges of our interest.

The simplified MRG equation, Eq. (3) thus represents an appropriate basis for qualitative estimations of the transport properties of negatively buoyant rigid microplastics in the water column. Note that rigidity of the particles is an essential condition which is why the advection of microfibers is out of the scope of this paper.

The connection between Eq. (3) and its approximation Eq. (1) is made by noticing that $\tau_\eta \approx 1s$ in the ocean (Monroy et al., 2017; Jiménez, 1997), so that the Stokes number $St = \tau_p/\tau_\eta$, which measures the importance of particle inertia in a turbulent flow, is very small (of the order of $10^{-3} - 10^{-2}$). Thus an expansion of the MRG equation for small $St$ (smallness of the Froude number, i.e. smallness of fluid accelerations with respect to gravity, is also required) can be performed. The expansion in its simplest form leads to (Balachandar and Eaton, 2010; Monroy et al., 2017; Drótos et al., 2019):

$$\boldsymbol{v} \approx \boldsymbol{u} + \boldsymbol{v}_s + \tau_p(\beta - 1)\frac{D\boldsymbol{u}}{Dt}. \tag{4}$$

We can now take the results of Monroy et al. (2017) for biogenic particles of sizes and densities similar to the microplastics considered here to show that the inertial corrections (the term proportional to $\tau_p$) in Eq. (4) are negligible, so that the simpler Eq. (1) correctly describes sinking of microplastics in the considered parameter range. For completeness, we report in App. B the explicit numerical calculations showing this (in which the influence of the Coriolis force is also taken into account, since it is known to be of the same order or larger than the inertial term when a large-scale flow is used for $\boldsymbol{u}$). In particular we find from release experiments from $1 \ m$ below the surface of a large number of particles with $\beta$ in the range $0.8 - 1$ in the whole Mediterranean that the difference between horizontal particle positions after 10 days of integration calculated from Eq. (4) and

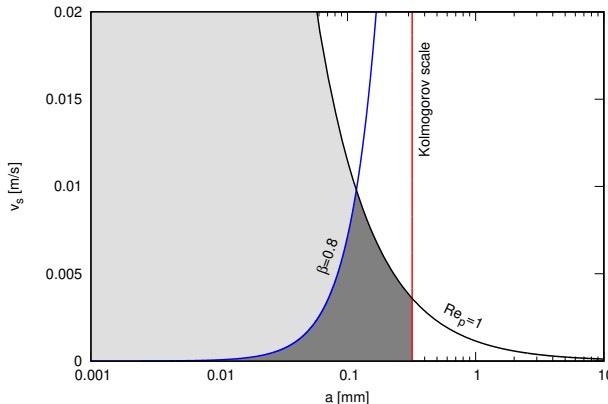

**Figure 2.** Settling velocities and particle sizes for which Eq. (3) holds. Kolmogorov scale is represented by the red line and $Re_p = 1$ with a black line, which bound the area of validity. Blue curve corresponds to $v_s = v_s(\beta, a)$ for $\beta = 0.8$, the upper bound to $v_s$ for typical microplastic densities. Dark shading denotes the plastic particle sizes and corresponding settling velocities for which application of Eq. (3) is valid.

the simpler Eq. (1) is just a 0.26% of the horizontal displacements. For the vertical motions the difference is of about 0.05%. Thus, Eq. (1) provides a proper description of the dynamics.

Even if an equation of motion is accurate, the accuracy of its solution is limited by that of the input data. In particular, small-scale flow features are absent from oceanic velocity fields $u$ simulated on large-scale domains, which is an important limitation of the respective solutions of Eq. (1). The NEMO velocity field of our choice is not an exception, but a rigorous evaluation of the corresponding errors of particle trajectories is not possible without knowing the actual small-scale flow. Nevertheless, one can roughly estimate the effect of these small scales by adding a stochastic term to Eq. (1) with statistical properties similar

to the expected ones for a small-scale flow (Monroy et al., 2017; Kaandorp et al., 2020). Results similar to those by Monroy et al. (2017), summarized in App. B, indicate that after 10 days of integration the relative difference between particle positions given by Eq. (4) with and without this 'noise' term modeling small scales (using $\beta = 0.99$) is around 8% for the horizontal displacements and 5% for the vertical ones. The figures become 12% and 5%, respectively, when evolving the particles for 20 days. These errors are moderate, although they may be of importance under some circumstances (Nooteboom et al., 2020). We

consider these figures as a baseline to evaluate corrections to the simple Eq. (1): adding more complex particle-dynamics terms to it will not improve plastic-sedimentation modeling unless the effect of these corrections is significantly larger than the above estimations for the effect of the unknown small-scale flow. In the following we consider the simple Eq. (1), but we estimate the implications of assuming or not a constant value of the water density.

## 4.2 Effect of variable seawater density

In this section we analyze the role of a variable seawater density on the particle settling dynamics. Fluid density is calculated from the TEOS-10 equations, which is a thermodynamically consistent description of seawater properties derived from a Gibbs function, for which absolute salinity is used to describe salinity of seawater and conservative temperature replaces potential temperature (Pawlowicz, 2010). In the simulations described in this section, as particles move in the ocean they encounter different temperatures and salinities, as given by the NEMO model described in Sect. 3.4, and then they experience different values of the ambient-fluid density.

We consider particles of a fixed density $\rho_p = 1041.5\ kg/m^3$. This implies that for a nominal water density of $\rho_f = 1025\ kg/m^3$ the value of $\beta$ would be $\beta = 0.99$, giving a sinking velocity $v_s = 6.2\ m/day$ for our particles of radius $a = 0.05\ mm$, but this sinking velocity will be increased or decreased in places where water density is lower or higher, respectively, so that we have a spatially- and temporally-dependent velocity in Eq. (1). The particle density and size have been chosen to be representative of the slowly-sinking microplastic particles, for which we expect the seawater density variations to have the largest impact. In this way we find some upper bound for the importance of variability in seawater density for particle trajectories.

We release $N = 78,803$ particles over the whole Mediterranean Sea, and monitor their trajectories under Eq. (1). The left panel of Fig. 3 shows the histogram of water densities encountered by the particles when the release is performed on July 8th 2000. On this summer date, the Mediterranean is well stratified, at least in its upper layers. Initially the particles are in surface waters with a range of salinities that average approximately to the nominal $\rho_f = 1025\ kg/m^3$. But as they sink in time they reach layers with higher densities (and more homogeneous across the Mediterranean). When the release is done in winter (right panel of Fig. 3) the water column is more mixed, so that the range of water densities experienced by the particles released at different points is always narrow. But the mean water density turns out to be always larger than the conventional surface density of $\rho_f = 1025\ kg/m^3$, so that a slightly slower sinking is expected to occur.

We illustrate the impact of this variable density on particle trajectories for the summer release in Fig 4. Here we compute, as a function of time, the range of horizontal $|\mathbf{x}^{(0)} - \mathbf{x}^{(1)}|$ and vertical $|z^{(0)} - z^{(1)}|$ distances and its average among particles. Trajectories $x^{(0)}(t)$ and $x^{(1)}(t)$ are obtained with constant nominal fluid density ($1025\ kg/m^3$) and position-dependent fluid density, respectively, using the same release location and date 8 July 2000 in both cases. Particle density is fixed at $\rho_p = 1041.5\ kg/m^3$. The difference between the two calculations (and thus the error of considering that constant value for the density) should be compared to average horizontal and vertical displacements of $95\ km$ and $124\ m$, respectively, at $t = 20$ days. At that time, we thus find that the influence of variable fluid density on the dynamics is about $3\%$ for the horizontal movement and $6\%$ for the vertical displacement on average.

A summary of the average relative differences on horizontal and vertical particle positions between using the location-dependent seawater density and a nominal constant value $\rho_f = 1025 kg/m^3$, both in winter and summer periods, is displayed in Table 1. The relative error produced by assuming a constant density is larger in the vertical direction. It is also larger for the release in winter, but this is a consequence of taking a value for the reference density that is not representative of winter waters but is strongly biased (see Fig. 3, right). If using a reference value more appropriate for winter waters (say $\rho_f \approx 1027\ kg/m^3$)

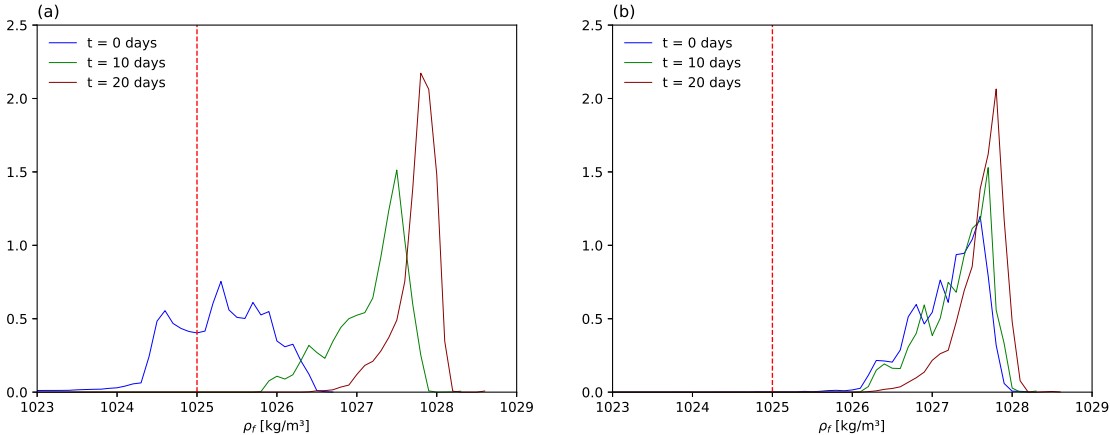

**Figure 3.** Normalized histogram of the seawater density $\rho_f$ at the positions of $N = 78803$ particles after $t = 0$, 10, and 20 days of being released over the whole Mediterranean. (a): summer release (release date 8 July 2000). (b): winter release (release date 8 January 2000). The particles' density is fixed at $\rho_p = 1041.5 kg/m^3$, and fluid density is obtained from the TEOS-10 equations. The vertical line indicates a conventional seawater density of $1025 kg/m^3$.

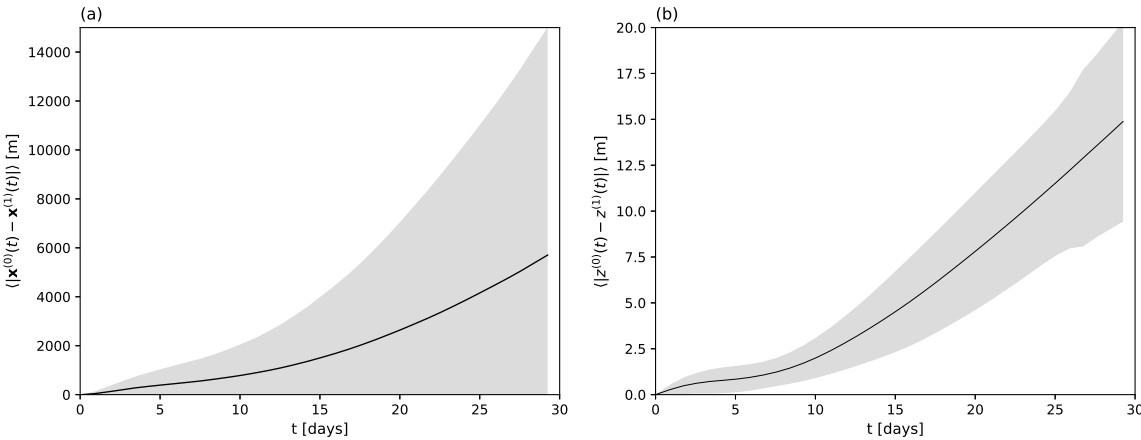

**Figure 4.** The distance, as a function of time, between trajectories obtained with constant nominal fluid density of $\rho_f = 1025 kg/m^3$ and the actual variable fluid density, both starting at the same initial location. The range of the values among all particles released in different points of the Mediterranean is indicated by the shaded area, while the solid line indicates the average over the particles. Particles have $\rho_p = 1041.5 kg/m^3$, and all parameters are the same as for the summer release in Fig. 3.(a): horizontal distances; (b): vertical distances.

the relative error remains quite small, due to the weaker stratification of the sea during this season. In fact, the reference value is also biased in the summer unless the investigation is restricted to the surface.

**Table 1.** Relative effect on horizontal and vertical particle positions after 10 and 20 days of integration, averaged over 78803 particles released over the whole Mediterranean at $1\ m$ depth, of replacing the actual seawater density by a nominal value $\rho_f = 1025\ kg/m^3$.

|  |  | 10 days | 20 days |
|---|---|---|---|
| Summer release | Horizontal: | 1.12 % | 2.75 % |
|  | Vertical: | 3.19 % | 6.25 % |
| Winter release | Horizontal: | 1.88 % | 5.62 % |
|  | Vertical: | 8.14 % | 9.32 % |

In brief, we see that the effect of location-dependent density may be a relevant effect to evaluate microplastic transport. At least, the traditional value of seawater density may be biased, which may be reflected in the particle trajectories. We recall, however, that we used parameters for the particle properties for which they are slowly falling. The impact of variable density on particles that sink faster will be smaller. Also, the effects reported in Table 1 remain of the order of the estimations of the effects of unresolved small scales of the flow (Sect. 4.1). As a consequence, in the following we will not consider variable seawater density, but restrict our modeling to Eq. (1) with a constant nominal value of the sinking velocity $v_s$.

### 4.3 Total mass and vertical distribution of microplastics

We will first estimate the total mass of negatively buoyant rigid microplastics in the water column of the open Mediterranean Sea by assuming a uniform vertical distribution, then we will justify this assumption by running numerical simulations according to the conclusion of Section 4.1 about the equation of motion.

For estimating the total mass, we take the quantities of Section 3.2 (4000 $tonnes/year$ of plastic release, with 37% being negatively buoyant of which 6% originates from maritime activities) to compute the rate $r$ at which microplastic particles of our interest enter the water column in the open sea: $r = 4000\ tonnes/year \times 0.37 \times 0.06 = 88.8\ tonnes/year$, or $r = 0.24\ tonnes/day$.

The next step is to estimate the time during which these microplastic particles remain in the water column before reaching the sea bottom. We take the mean depth for the Mediterranean to be $h = 1480\ m$ (Eakins and Sharman, 2010; GEBCO Compilation Group, 2020) and estimate a residence time $\tau$ as the time of sinking to that mean depth. The residence time depends on the sinking velocity, $\tau = h/v_s$, and thus on the physical properties of the microplastic particles. Assuming a seawater density $\rho_f = 1025\ kg/m^3$, and the range of plastic densities and their proportions described in Sect. 3.1, we see from Eq. (2) that for microplastic particles of radius $a \approx 0.05\ mm$ the range of sinking velocities is $6.20 - 509.23\ m/day$, giving a residence time in the range $3.1 - 255\ days$. Averaging these times weighted by the proportion of each type of plastic we get $\overline{\tau} \approx 14\ days$. Combining the input rate $r$ with this mean residence time we get an estimate for the total amount present in the water column at any given time as $Q = r\overline{\tau}$: the result is $Q \approx 3.36$ tonnes of dense rigid microplastics if all of them would be in the form of

particles of size $a = 0.05\ mm$. This is below but close to 1% of the estimated upper bound of 470 tonnes of floating plastic in the Mediterranean (according to the corresponding estimation of Kaandorp et al. (2020)).

We emphasize the many uncertainties affecting this result (Sections 3.1 and 3.2), and we highlight the one related to particle size: because of the quadratic dependence of the sinking velocity on the particle radius $a$, Eq. (2), choosing the particle size to be half of the one used here will lead to a four times larger estimate for the mass if the same release rate is assumed. This enhanced retention of smaller particles in the water column may imply, depending on the actual size distribution, a dominance of very small particles on the plastic mass content of the water column. However, our estimates of plastic input into the ocean (we use

mainly Kaandorp et al. (2020)) rely on observations that do not catch extremely small particles. These considerations further justify our choice of a radius $a = 0.05\ mm$, small but still easily detectable, as convenient to provide reasonable estimations of negatively buoyant rigid microplastic mass in the water column within commonly quoted size ranges. We can not exclude larger plastic content at smaller sizes. Another source of bias may be not considering in this study the impact of small-scale turbulence and convective mixing events. While small-scale turbulence might cause an increase of lifetimes of particles in the

water column, dense water formation and rapid convection, a process reported in areas such as the Gulf of Lions, might likely reduce particle retention time. These events take place in winter and were shown to transfer particles from the ocean surface to mid-waters (1000 meters) and deep ocean (>2000 meters) in a very short time (1-2 days) and lead to the formation of bottom nepheloid layers (de Madron et al., 1999; Vidal et al., 2009; Heussner et al., 2006; Stabholz et al., 2013).

  The result for the total mass is independent of the horizontal distribution of particle release, which is quite inhomogeneous

(Fig. 1 of Liubartseva et al., 2018). However, for a rough estimate of the density of these microplastics in the water column, we assume a uniform particle distribution over the whole Mediterranean both in horizontal and in vertical. Since the volume of the Mediterranean is about $4.39 \times 10^6\ km^3$ (Eakins and Sharman, 2010) the estimated density would be $\rho_V \approx 7.7 \times 10^{-11}\ kg/m^3$ (with the above-discussed scaling issues with $a$). We remind the reader that this is a value for the open sea, and our study does not address coastal areas, where the density would likely be higher.

The above estimates are rather rough as a result of the mentioned uncertainties. The assumption of a uniform distribution in the vertical direction has not yet been justified either, but we will show it to be appropriate by means of our simulations of particle release starting at $1\ m$ depth over the whole Mediterranean. Instead of performing a continuous release of new particles at each time step, and computing statistics over this growing number of sinking particles, we approximate this by the statistics of all positions at all time steps of a set of particles deployed in a single release event. This assumes a time-independent fluid

flow, but this approximation is appropriate, since the dispersion of an ensemble of particles released in a single event follows rather well-defined statistical laws, see Section 4.4, and is thus independent of the time-varying details of the flow. Particles are removed when touching the bottom. For our estimate, we use $\beta = 0.8$, i.e., assuming the fastest sinking velocity of typical plastic particles, for which particles reach vertical depths deeper if compared with the slower sinking velocities used in our study.

Figure 5 shows $\rho(z)$, the density of plastic particles per unit of depth $z$ in the whole Mediterranean, and also $A(z)$, the amount of area that the Mediterranean has at each depth $z$. Both functions have been normalized such that the value of their integrals with respect to $z$ is one; in this way the functions can be displayed in the same plot. We see that both curves are

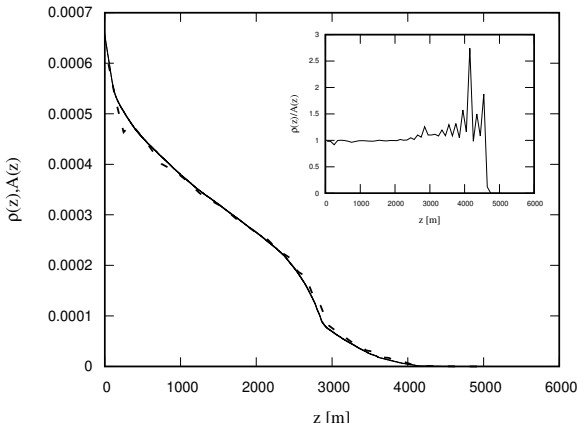

**Figure 5.** The continuous line is the area that the Mediterranean has at each depth $z$. The dashed line is microplastic density per unit of depth $\rho(z)$ under continuous release of particles with $\beta = 0.8$ at $1\ m$ depth. Both curves have been normalized to have unit area, so that they can be compared on the same scale. The binning size is $100\ m$. The inset shows the ratio $\rho(z)/A(z)$, proportional to the mass density of microplastic per unit of volume $\rho_V(z)$.

nearly identical (in fact, just proportional, because of the normalization), indicating that the variation of the number of particles with depth is essentially due to the decrease of sea area with depth. A clearer way to see that is to plot $\rho(z)/A(z)$, which is proportional to the mean plastic concentration per unit volume at each depth $z$, $\rho_V(z)$. We see that this quantity is nearly constant, at least in the first 3000 m. At larger depths a weak increase seems to occur, but made unclear by the poor statistics arising from the small area and number of particles present at these depths. Thus, the hypothesis of a uniform distribution of plastic in the water column seems to be a reasonable description of the simulation of the fastest-sinking particles.

A uniform distribution of plastics in $z$ is what is expected if the vertical velocity of the particles is exactly a constant (since each particle will spend exactly the same time at each depth interval). The equation of motion used, Eq. (1) corrects this constant sinking velocity $\boldsymbol{v_s}$ with a contribution $\boldsymbol{u}$ from the ambient flow. Thus, the close-to-constant character of the plastic concentration may imply that the flow correction $\boldsymbol{u}$ is negligible, at least when considering its effect over the whole Mediterranean. Another possibility is that the fluctuating flow component $\boldsymbol{u}$ in Eq. (1) results in a vertical dispersion compatible with a constant concentration. Although the former explanation predicts an alteration from a constant if the settling velocity is sufficiently small to allow $\boldsymbol{u}$ to induce a stronger vertical dispersion, we will see in the next section that a nearly constant concentration may be assumed for the majority of our parameter range.

## 4.4 Transient evolution

We now analyze in detail the transient evolution of particle clouds initialized by flash releases at a fixed depth. Numerically we proceed by releasing $N = 78803$ particles uniformly distributed over the entire Mediterranean surface at $1\ m$ depth in the winter season, as already described. They evolve according to Eq. (1) using a constant water density. We take three examples for

the particle density, which correspond to $\beta = 0.8, 0.9, 0.99$, or $v_s = 153.48, 68.21, 6.20$ m/day for our particles of radius $a = 0.05\ mm$, respectively; in what follows, these setups shall be denominated as v153, v68 and v6. The horizontal displacements during the particle sinking times are much larger (of the order of $60\ km$) than the sea depth, so that in fact the particles are sinking *sideways* (Siegel and Deuser, 1997). However, the horizontal displacements still remain very small compared to the basin size, and we concentrate on the vertical motion. Even though the vertical steady distribution has been found to be close to uniform in Section 4.3, the reason for this is not evident, and we will give support here for the pertinence of this finding to most of the relevant parameter range.

Figure 6 shows the vertical particle distribution at different times (upper plot is for v153, middle for v68 and bottom for v6). The plot is given in terms of a rescaled variable $\tilde{z} = \frac{z - t v_s}{\sigma_z}$ where $\sigma_z^2 \equiv \langle (z_i - \langle z_i \rangle)^2 \rangle$ is the variance of the particles' $z$ coordinate. Here the subindex $i$ refers to the particle and $\langle \dots \rangle$ denotes averaging over different particles. Thus, we plot in the figure the rescaled distribution of the particles around the average depth of the particles at any given time. For comparison, the normal distribution is plotted with dashed lines. This figure shows deviations from Gaussianity for early times. The deviation from normal distribution decreases for later instants but remains considerable, especially for the tails, which may also be indicative of anomalous diffusive behavior. For reference, particles reach the mean Mediterranean depth, $h = 1480\ m$ at times $\tau = 9.64, 21.8$ and $246.7$ days for v153, v68 and v6, respectively.

Since a non-Gaussian distribution is usually linked to anomalous dispersion (Neufeld and Hernández-García, 2009), we now analyze this aspect in detail by considering how the variance of the vertical particle distribution, $\sigma_z^2(t)$, evolves. Although there is a continual loss of particles because of reaching the seafloor with a varied topography, we illustrate in App. C that our conclusions are likely unaffected by this effect.

According to Fig. 7, dispersion appears to be governed by different laws in different regimes, which we shall distinguish by the approximate effective exponents $\nu$, defined through approximate behaviors $\sigma_z^2 \sim t^\nu$ in different time intervals.

We start our analysis with the fastest-sinking particles (v153, Fig. 7a). At the very beginning, superdiffusion takes place with $\nu > 2$, which may be related to autocorrelation in the flow, but we will iterate on this question when comparing different settling velocities. Around $t = 1$ day, the evolution seems to become consistent with normal diffusion ($\nu = 1$), usual after initial transients in oceanic turbulence (Berloff and McWilliams, 2002; Reynolds, 2002). However, around $t = 4.5$ days, we can observe a crossover to ballistic dispersion ($\nu = 2$).

We explain this last crossover as resulting from a different mean sinking velocity in diverse regions of the Mediterranean, associated with up- and down-welling. This can be modeled in an effective way by writing the vertical position of particle $i$ as

$$z_i = \langle z_i \rangle + \bar{\omega}_i t + W_i, \tag{5}$$

where $\langle \dots \rangle$ denotes, as before, an averaging over different particles. Here we are assuming that $z_i - \langle z_i \rangle$ evolves according to the sum of a constant average *velocity* contribution $\bar{\omega}_i$ for sufficiently long times (a characteristic of the flow region traversed by particle $i$), and of $W_i$, a Wiener process representing fluctuations with zero mean and defining a diffusion coefficient $D_i$ for each trajectory by $\overline{W_i^2} = D_i t$. The overbar refers to temporal averaging for asymptotically long times along the trajectory of a given particle (but assuming that the particle remains in a region with a well-defined $\bar{\omega}_i \neq 0$), and $D_i$ characterizes the strength

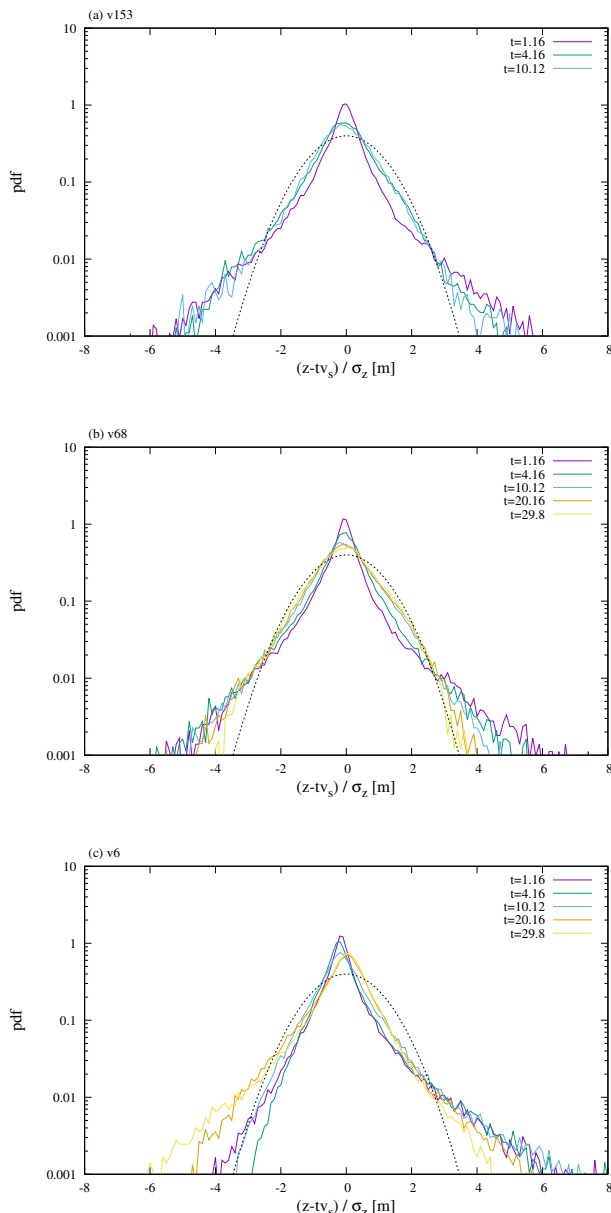

**Figure 6.** The probability density function, estimated from a histogram of bin size $0.1$, of all particles released in the Mediterranean in the rescaled variable $\tilde{z} = \frac{z - tv_s}{\sigma_z}$ for the different setups (v153, v68 and v6) and times (in days) as indicated. For comparison, normal distributions of zero mean and unit variance are shown with a dashed line.

of the fluctuations. Assuming $\langle \bar{\omega}_i W_i \rangle = 0$,

$$\sigma_z^2 \equiv \langle (z_i - \langle z_i \rangle)^2 \rangle = \langle \bar{\omega}_i^2 \rangle t^2 + \langle D_i \rangle t, \tag{6}$$

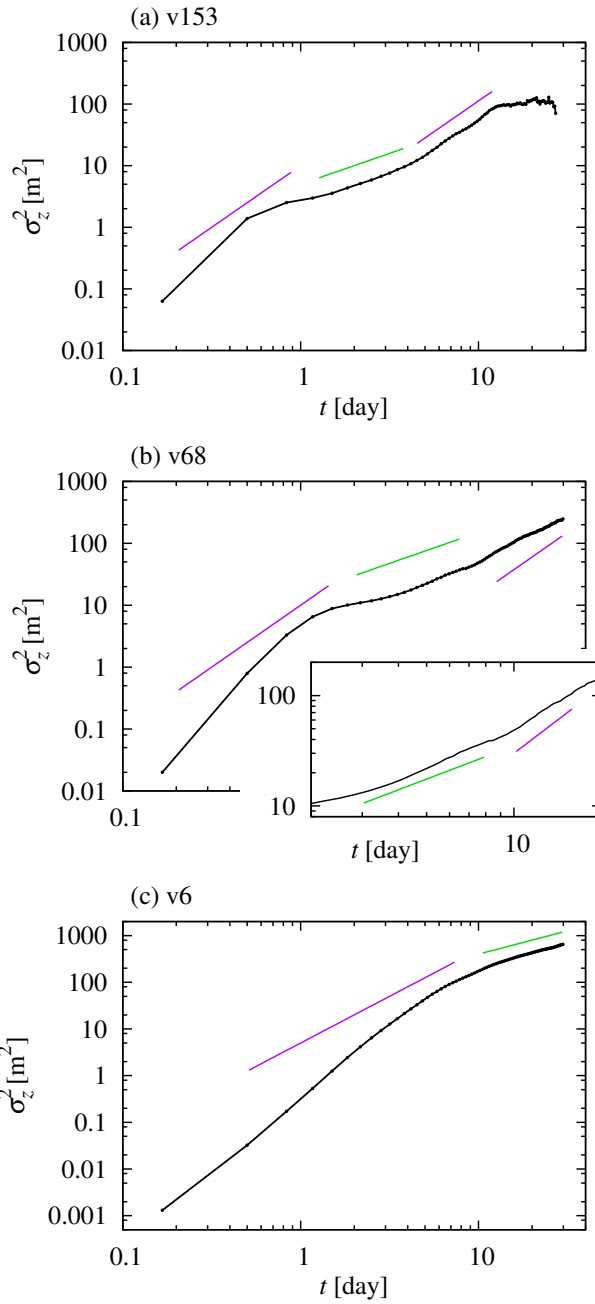

**Figure 7.** Variance of depth reached by the particles as a function of time. Straight lines represent power laws for reference, with exponents 1 (in green, corresponding to standard diffusion) and 2 (in purple, corresponding to ballistic dispersion).

that is, the variance is a sum of a ballistic and a normal diffusive term, associated with regional differences in the mean velocity and with fluctuations, and dominating for long and short times, respectively. Writing $D = \langle D_i \rangle$, the crossover between the two

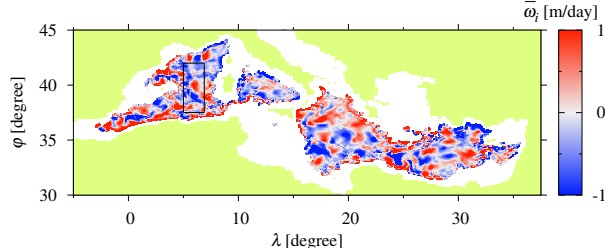

**Figure 8.** $\bar{\omega}_i$ as estimated from $t = 10.83$ days plotted at the initial position of each particle $i$ in the v153 simulation. The black rectangle in the Western Mediterranean is the area of large depth considered in App. C.

regimes is obtained by equating the two terms as

$$t^* = \frac{D}{\langle \bar{\omega}_i^2 \rangle}. \tag{7}$$

To evaluate Eq. (7), we first estimate $\bar{\omega}_i$ for each particle from the "asymptotically" long time of $t = 10.83$ days, which is the latest time after the crossover still in the ballistic regime in Fig. 7a, when the contribution of fluctuations should already have become negligible. The horizontal pattern of the estimated $\bar{\omega}_i$ is presented in Fig. 8, which confirms its patchiness throughout the Mediterranean, associated with mesoscale features. Computing $\langle \bar{\omega}_i^2 \rangle$ and fitting a line to $\sigma_z^2(t)$ between $t = 1.4$ and $4$ days to estimate $D$, we obtain $t^* \approx 4.5$ days from Eq. (7), which remarkably agrees with Fig. 7a. After approximately $t = 12$ days there is hardly any dispersion, since most of the particles are close to the sea bottom (cf. Fig. 9) where the vertical fluid velocity is nearly zero. Note also a small drop in $\sigma_z^2($ at the very end of the time series, where the results may actually be subject to artifacts, see App. C. However, this is of minor importance, since the distribution of particles so close to the bottom should anyway be strongly influenced by resuspension and remixing by bottom currents (Kane et al., 2020).

The different regimes are not as clear in the v68 case as for v153, see Fig. 7b. One evident novel feature is a subdiffusive regime during the transient from the initial superdiffusion (as in the case of horizontal tracer dispersion in the ocean studied by Berloff and McWilliams (2002); Reynolds (2002)). Approximate normal diffusion is then observed until $t = 10$ days, when a crossover to a faster dispersion does seem to take place, see the inset. A fit of normal diffusion from $t = 4$ to $8$ days and the velocity variance at $t = 12.5$ days give an estimate $t^* \approx 11.7$. However, the long-time ballistic regime is not clear. In fact, a long-term return from such a ballistic regime to normal diffusion is expected as a result of increasing horizontal mixing, which renders $\bar{\omega}_i$ time dependent and makes it approach zero. According to a careful visual inspection of the inset in Fig. 7b, this may take place already around $t = 14$ days.

For v6 (Fig. 7c), the transition from the initial superdiffusive regime to that of normal diffusion appears to not involve subdiffusion. This is already informative: the fluid velocity field is the same for the three simulations with different settling velocity, so the differences must originate from the different rate of sampling of the different fluid layers by particles while they sink. In particular, the decay of autocorrelation is obviously faster for faster-sinking particles, since it is determined by the spatial structure of the velocity field.

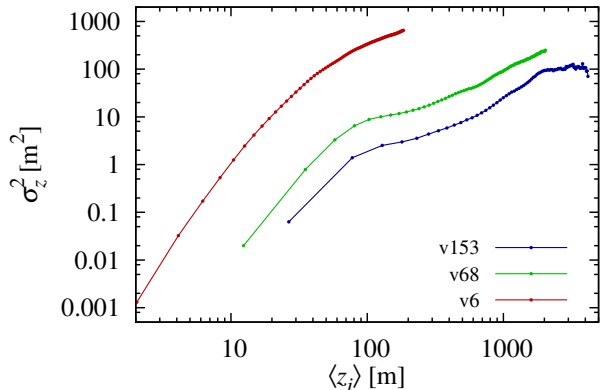

**Figure 9.** Variance of depth reached by the particles as a function of their mean depth.

While this is one possible explanation for the earlier timing of the initial transition from anomalous to normal diffusion for higher settling velocity, one cannot exclude that a depth-dependent organization of the flow is more in play; note that $\nu > 2$ at the beginning, which might not be explained by simple autocorrelation but might be characteristic of properties of the velocity field at those depths. The governing role of the spatial structure is supported by Fig. 9: the transition in question takes place at the same depth ($\approx 100$m) in the different simulations, which seems to point to mixed-layer processes. Depth-dependence might also govern the suppression of ballistic dispersion for long times, but it is very unclear.

Note in Fig. 9 that the vertical variance is not expected to grow much larger for v68 than for v153 even if the simulation were longer. Therefore, even if the constancy of the steady vertical distribution relies on the weak vertical dispersion for v153 (see Section 4.3), constancy is expected to hold in most of our parameter range. A considerably stronger dispersion and a possible corresponding deviation from constancy may arise only for extremely low settling velocities, like for v6 in Fig. 9.

## 5 Conclusions

We have discussed the different types of plastics occurring in the water column, pointing out gaps in our knowledge about the sources, transport pathways and properties of such particles. It would be highly beneficial to have distributions of size, polymer type and quantifiers of shape recorded separately for the dynamically different classes of microplastics.

We have focused our attention on rigid microplastic particles with negative buoyancy. We have argued that the simplified MRG equation approximates the dynamics of such particles sufficiently well for qualitative estimations.

We have then analyzed the importance of different effects in this equation, and concluded that the Coriolis and the inertial terms are negligible. When a velocity field of large-scale nature is input to the equation (such that small-scale turbulence is not resolved), or when the variability in seawater density is neglected, moderate but possibly non-negligible errors emerge (Nooteboom et al., 2020). However, our conclusions about the vertical distribution and dispersion of microplastics rely on

robust features of the large-scale flow and must remain unaffected by moderate errors. We also note that the traditional value of seawater density, $\rho_f = 1025 \ kg/m^3$, is representative only for near-surface layers in the summer, and correcting for the bias could reduce the error of simulations with a constant seawater density. A suitable equation of motion for the particles considered is constructed by adding to the the external velocity field a constant settling term, as also found by Monroy et al. (2017) for marine biogenic particles.

When the velocity field of the Mediterranean sea is approximated by realistic simulation, this equation of motion results in a nearly uniform steady distribution along the water column, perhaps except at extremely low settling velocities. The corresponding total amount of plastic present in the water column is relatively small, close to 1% of the floating plastic mass, but it may be an important contribution to the microplastic pollution in deep layers of the ocean, and is subject to several uncertainties.

Note that only those microplastic particles are considered here that have not yet sedimented on the bottom, and the plastic amount sedimented on the seafloor is large (Fischer et al., 2015; Liubartseva et al., 2018; Peng et al., 2018; Mountford and Morales Maqueda, 2019; Soto-Navarro et al., 2020). The suitability of our equation of motion to describe the sinking of a class of microplastic particles implies that advection by the flow may contribute to large-scale horizontal inhomogeneity of deep-sea plastic sediments by means of recently described noninertial mechanisms (Drótos et al., 2019; Monroy et al., 2019; Sozza et al., 2020). This may especially be so in regions where redistribution by bottom flows is restricted to small distances, like abyssal plains (Kane and Clare, 2019). Resuspension and redistribution may be dominant in forming sedimented patterns (Kane et al., 2020), and a future investigation should take all processes into account to identify zones of high plastic concentration on the sea bottom.

As for the vertical distribution profile, its approximate uniformity may be linked to the weak vertical dispersion of particles that is found in our simulations started with a flash release over the whole surface of the Mediterranean sea. The shape of the emerging transient vertical distribution exhibits deviations from a Gaussian, which are related to anomalous diffusive laws that dominate the vertical dispersion process in some phases.

The different diffusive laws are related to the properties of the decay in the Lagrangian velocity autocorrelation defined along the trajectories of the sinking particles. An important example is the transition from initial superdiffusion to a longer phase of normal diffusion, occurring around 100 m depth, which indicates that the particles enter into a different flow regime. Another characteristic of the velocity field is a horizontal patchiness, which results in a long-term ballistic dispersion as long as the particles' horizontal displacements remain small. The vertical diffusion returns to the normal type when horizontal mixing becomes more developed. These results suggest regional differences in the sinking process, so that regional modeling might be more appropriate than a whole-basin approach. Future studies will include different areas of the oceans, and analyze the role of Lagrangian coherent structures on the different vertical dispersion regimes.

*Data availability.* The velocity field from the NEMO simulation used in our study can be downloaded from http://opendap4gws.jasmin.ac.uk/thredds/nemo/root/nemo_scan_catalog.html. Parcels is a set of Python classes developed under the TOPIOS project and is accessible at

https://oceanparcels.org/. The scripts for running the particle transport simulations are available upon request from Rebeca de la Fuente, rebeca@ifisc.uib-csic.es.

## Appendix A:  Deviations from a spherical particle shape

We quantitatively assess the impact of deviations from a spherical shape through a correction to the settling velocity $v_s$. The simplified MRG equation, Eq. (3), or its first-order approximations in the Stokes number, Eqs. (4) and (B1), are affected by
particle geometry through the drag force and the added mass term; however, accelerations are irrelevant for $v_s$, so that the added mass term does not appear in its formulation or in the simple approximation of Eq. (1). We will compare values of the settling velocity describing nonspherical and spherical particles with the same density, then finally comment on the results' relevance for Eqs. (3), (4) and (B1).

Most generally, the settling velocity vector $\mathbf{v}_s$ can be obtained by balancing the drag force $\mathbf{F}_{\text{drag}}(\mathbf{v} - \mathbf{u})$ (a function of the
difference of the particle and the fluid velocities, $\mathbf{v}$ and $\mathbf{u}$, respectively) with the resultant of gravitational and buoyancy forces:

$$0 = \mathbf{F}_{\text{drag}}(\mathbf{v} - \mathbf{u}) + V \left(\rho_{\text{p}} - \rho_{\text{f}}\right) \mathbf{g} \tag{A1}$$

with $\mathbf{v} - \mathbf{u} = \mathbf{v}_s$, where $V$ is the particle's volume, $\rho_{\text{p}}$ and $\rho_{\text{f}}$ are the densities of the particle and the fluid, respectively, and $\mathbf{g}$ is the gravitational acceleration vector. For a spherical particle with radius $a$, the Stokes drag force reads as

$$\mathbf{F}_{\text{drag}}^{(\text{sph})}(\mathbf{v} - \mathbf{u}) = -6\pi\mu(\mathbf{v} - \mathbf{u})a, \tag{A2}$$

where $\mu$ is the dynamical viscosity of the fluid. According to Leith (1987); Ganser (1993), an appropriate approximation for small nonspherical particles is

$$\mathbf{F}_{\text{drag}}^{(\text{non})}(\mathbf{v} - \mathbf{u}) = -6\pi\mu(\mathbf{v} - \mathbf{u}) \left(\frac{1}{3}a_{\text{n}} + \frac{2}{3}a_{\text{s}}\right), \tag{A3}$$

where $a_{\text{n}}$ is the radius of the sphere with equivalent area projected on the plane perpendicular to the relative velocity $\mathbf{v} - \mathbf{u}$,
and $a_{\text{s}}$ is the radius of the sphere with equivalent total surface. From either of the last two equations, the settling velocity is obtained by substituting $\mathbf{v} - \mathbf{u} = \mathbf{v}_s$, and solving Eq. (A1) for $\mathbf{v}_s$. We denote the magnitudes of the settling velocities obtained from Eq. (A2) and Eq. (A3) by $v_s^{(\text{sph})}$ and $v_s^{(\text{non})}$, respectively.

To characterize the correction in the settling velocity for a given nonspherical particle (with a given density $\rho_{\text{p}}$) with respect to assuming a spherical shape with a radius $a$, we will consider

$$q \equiv \frac{v_s^{(\text{non})}}{v_s^{(\text{sph})}} = \frac{3}{4\pi} \frac{V^{(\text{non})}}{a^2 \left(\frac{1}{3}a_{\text{n}} + \frac{2}{3}a_{\text{s}}\right)}, \tag{A4}$$

where $V^{(\text{non})}$ is the real volume of the given particle. In order to evaluate Eq. (A4), one has to specify the shape and the size of the particle, its orientation with respect to its relative velocity, and also how $a$ is derived from its real size.

Note that it is always possible to define an $a$ for which $q = 1$, i.e., for which there is no correction arising from the deviation from a spherical shape. In this sense, any choice of $a$ representing a spherical shape, including ours in the manuscript, describes the settling velocity of certain nonspherical particles, the question is just their shape and size, which will mutually depend on each other for a given $a$. We will nevertheless proceed by choosing a shape class and defining $a$ along independent considerations, because we intend to link a given $a$ to a single particle size as identified during the processing of field observations.

The shape of rigid microplastic particles is not usually described in the literature, but we can see photographs of some examples in, e.g., Song et al. (2014); Fischer et al. (2015); Bagaev et al. (2017). For an explorative computation, a reasonable choice seems to be a rectangular cuboid with edges $A$, $B = \hat{B}A < A$ and $C = \hat{C}A < B < A$, where one or both of $\hat{B}$ and $\hat{C}$ are less than 1 but greater than, say, $0.1$.

Under this assumption, the particle size will correspond to the longest edge, $A$, of the cuboid if the size is identified through microscopy as the largest extension ("length"; e.g., Cózar et al., 2015); and it may be related more to the middle edge, $B$, if one thinks of a sieving technique (e.g., Suaria et al., 2016). The naive choice will be $a = A/2$ and $a = B/2$ in these two cases.

We can substitute either of these choices of $a$ in Eq. (A4), as well as the appropriate formulae describing the actual cuboid. $V = ABC$ is unique, and so is $a_{\mathrm{s}}$,

$$a_{\mathrm{s}} = \left( \frac{AB + AC + BC}{2\pi} \right)^{\frac{1}{2}}. \tag{A5}$$

However, $a_{\mathrm{n}}$ depends on the particle's orientation with respect to the relative velocity. Implications will be discussed when interpreting the results, and we take here all three directions parallel to edges $A$, $B$ and $C$ to represent different possibilities. The corresponding expressions for $a_{\mathrm{n}}$ read as

$$a_{\mathrm{n},X} = \left( \frac{ABC}{\pi X} \right)^{\frac{1}{2}} \tag{A6}$$

for $X = A$, $B$ and $C$.

After substituting all these expressions in Eq. (A4), we obtain

$$q_X^{(A/2)} = 9\pi^{-\frac{1}{2}} \hat{B}\hat{C} \left[ \left( \frac{\hat{B}\hat{C}}{\hat{X}} \right)^{\frac{1}{2}} + 2^{\frac{1}{2}} \left( \hat{B} + \hat{C} + \hat{B}\hat{C} \right)^{\frac{1}{2}} \right]^{-1}, \tag{A7}$$

$$q_X^{(B/2)} = \hat{B}^{-2} q_X^{(A/2)}, \tag{A8}$$

where $X = \hat{X}A$ has been introduced.

We plot $q_A^{(A/2)}$ in Fig. A1 as a function of $\hat{B}$ and $\hat{C}$ for $\hat{B}, \hat{C} \in [0.1, 1]$ with $\hat{B} > \hat{C}$. Its range extends from $0.07$ to $1.5$ on this domain, but it drops below $0.1$ only for $\hat{C}$ very close to $0.1$ and $\hat{B}$ below $0.2$; i.e., for extremely thin rod-like particles, which do not appear to be common based on photographs. The range of $q_A^{(B/2)}$ (not shown) on the same domain is between $0.2$ and $7$, and values above $4$ are again restricted to very small $\hat{C}$ and to $\hat{B} < 0.3$. The results are very similar for other choices of $X$, deviations beyond $20\%$ with respect to $X = A$ are only found for small $\hat{C}$ and do not reach beyond $40\%$ even there.

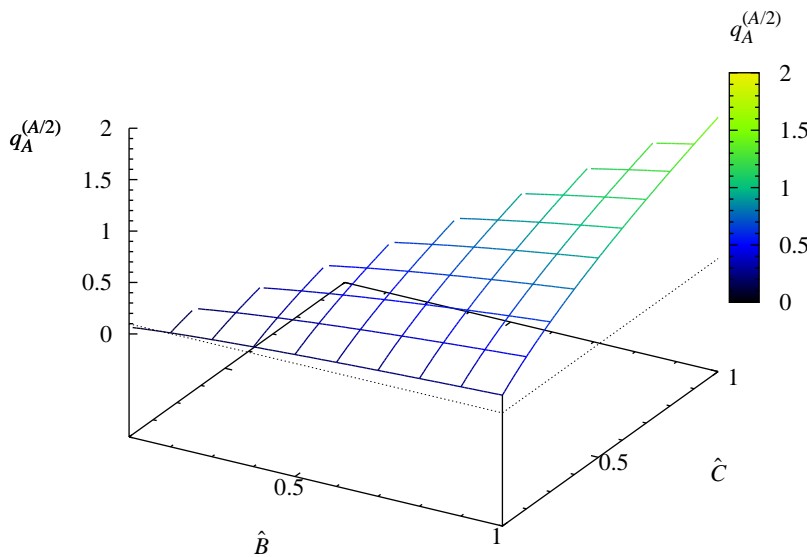

**Figure A1.** $q_A^{(A/2)}$ as a function of $\hat{B}$ and $\hat{C}$ for $\hat{B}, \hat{C} \in [0.1, 1]$ with $\hat{B} > \hat{C}$. Dotted lines represent $q_A^{(A/2)} = 0.1$.

We have left the question which orientation is relevant open so far. In small-scale isotropic turbulence, which is certainly present in the ocean, nonspherical particles have a preferential alignment with certain characteristics of the flow but undergo rotation (Voth and Soldati, 2017). This is why we have chosen to simply cover three perpendicular orientations in our analysis, and have found that differences that may arise from changes in the orientation are minor in most of the domain describing shapes. The only relevant exception is small $\hat{C}$ with $\hat{B} \approx 1$. This regime may characterize paint flakes (Song et al., 2014; Bagaev et al., 2017), but the relative difference remains below $40\%$ even there.

Even though the real advection of the particles will become more complicated as a result of the ever-changing orientation and may thus be beyond the scope of the MRG equation (cf. the discussion in the main text about the settling velocity of irregular particles), we have found that changing orientation introduces minor variations in the value of the settling velocity. Together with the absence of order-of-magnitude corrections that may arise from a nonspherical shape (but comparing shapes under the assumption of the same particle density), this gives quantitative support for the applicability of a spherical shape in Eq. (1) of the main text.

Finally, we briefly comment on the more general Eqs. (4) and (B1), in which effects from nonsphericity arise in the inertial term through added mass. Since corrections in added mass with respect to a sphere are of order $1$ for all common shapes (see Kaneko et al., 2014, for an overview), we believe that the finding of App. B about the negligible importance of inertial effects in these equations is not affected by a deviation from sphericity. The Stokes number, which is proportional to the settling

velocity and also depends on the coefficient of added mass ($St \sim \tau_p$ with $\tau_p$ given by Eq. (2)), is estimated to be $10^{-3} - 10^{-2}$

for spherical particles in Sect. 4.1, so that it will not increase to 1 due to a nonspherical shape either, hence leaving the approximation (4) of Eq. (3) valid.

## Appendix B: Importance of different physical effects in the dynamics

We present here the detailed numerical analysis of the relevance of a finite time of response (Stokes time, $\tau_p$) of the particle to the fluid forces, the Coriolis force, and scales unresolved by the NEMO velocity field.

We incorporate the first two effects to a single equation,

$$\mathbf{v} = \mathbf{u} + \mathbf{v}_s + \tau_p(\beta - 1)\left(\frac{D\mathbf{u}}{Dt} + 2\mathbf{\Omega} \times \mathbf{u}\right) , \tag{B1}$$

which is identical to Eq. (4) except for the addition of the Coriolis force, $2\mathbf{\Omega} \times \mathbf{u}$. $\mathbf{\Omega}$ is Earth's angular velocity vector. We include the Coriolis force because it can be more important than the other inertial term, given by the fluid acceleration $D\mathbf{u}/Dt$, in large-scale ocean flows $\mathbf{u}$ (Haller and Sapsis, 2008; Monroy et al., 2017).

The effect of unresolved scales will also be estimated by keeping the original NEMO velocity field $\mathbf{u}$ but modifying the equation of motion, Eq. (1), by adding a stochastic noise term:

$$\mathbf{v} = \mathbf{u} + \mathbf{v}_s + \mathbf{W} . \tag{B2}$$

$\mathbf{W}(t) = (\sqrt{2D_h}\xi_x(t), \sqrt{2D_h}\xi_y(t), \sqrt{2D_v}\xi_z(t))$, where $\boldsymbol{\xi}(t)$ is a vector Gaussian white noise process (independent for each particle) of zero mean and with correlations given by $\langle \xi_i(t_1)\xi_j(t_2) \rangle = \delta_{ij}\delta(t_1 - t_2)$ , $i, j = x, y, z$. Thus, the horizontal and

vertical intensities of this term are given by $D_h$ by $D_v$, respectively.

The statistical properties are chosen to be similar to the ones expected for oceanic motions below the scales resolved by the numerical model (Monroy et al., 2017; Kaandorp et al., 2020). To do so, we use for $D_h$ Okubo's empirical formulation (Okubo, 1971) that parameterizes the effective horizontal eddy-diffusion below a spatial scale $\ell$ as $D_h(\ell) = 2.055 \times 10^{-4}\ell^{1.55} m^2/s$. Taking for $\ell$ the horizontal resolution of our numerical model (1/12 degrees) we obtain $D_h = 7.25 \ m^2/s$. Since the Okubo

formula is an empirical fit to surface motions, and effective horizontal diffusivity should be weaker below the thermocline, our results provide an upper bound for the error associated with unresolved scales of fluid motion. For the vertical diffusivity we take $D_v = 10^{-5}m^2/s$.

In order to compare the different equations of motion, we release a large number $N = 78803$ of particles on the whole Mediterranean at $1 \ m$ depth on 8 January 2000. We associate a $\beta$ parameter to each particle by selecting it from a random

uniform distribution in the range $\beta \in [0.8, 1)$, and once it is selected it remains fixed at all times for the corresponding particle. High values of $\beta$ (close to one) correspond to more buoyant plastic particles whereas low values of $\beta$ correspond to high settling velocities. The corresponding range of velocities is $v_s \in [1.776 \times 10^{-3}, 0) \ m/s$. We integrate the particle trajectories using Eq. (1) and also, from the same initial conditions, using the corrected dynamics in Eq. (B1) or (B2), all with the same NEMO velocity field $\mathbf{u}$. The horizontal and vertical distances between particles released from the same point but evolved with

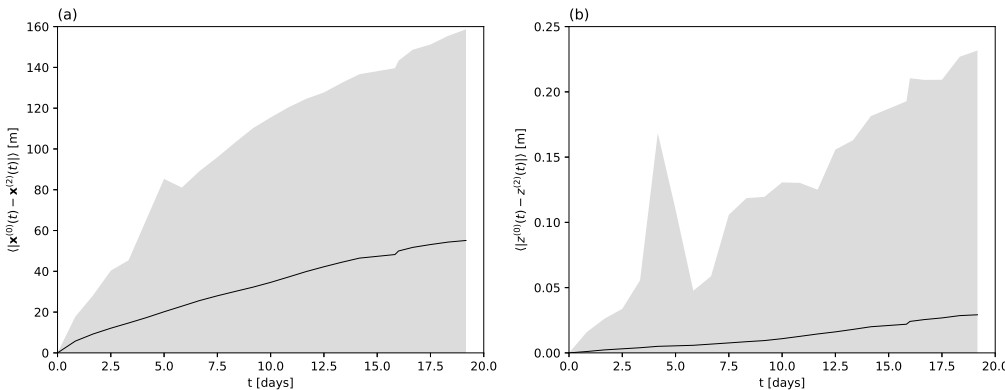

**Figure B1.** Solid line indicates the average horizontal distance $d_h^I$ (a) and the average vertical distance $d_v^I$ (b) of particles released from the same initial location but integrated with equations (1) and (B1) in the NEMO velocity field. Shaded region indicates the range of the distances among the individual pairs of particles.

different equations are compared by calculating the following averages over particles:

$$d_h^I(t) = \frac{1}{N} \sum_{k=1}^{N} |\mathbf{r}_k^0(t) - \mathbf{r}_k^I(t)| \ , \ d_v^I(t) = \frac{1}{N} \sum_{k=1}^{N} |z_k^0(t) - z_k^I(t)| \ . \tag{B3}$$

$\mathbf{r}_k = (x_k, y_k)$ is the horizontal position of particle $k$, $z_k$ is its vertical position, and the superindices $0$ and $I$ indicate that the particle trajectory has been integrated by using the reference equation, Eq. (1) or the one containing inertial corrections, Eq. (B1). The quantities $d_h^W(t)$ and $d_v^W(t)$, comparing Eq. (1) with the dynamics (B2) modeling small-scale flow effects, are

defined analogously.

    In Fig. B1, we display the average distances $d_h^I(t)$ and $d_v^I(t)$ as a function of time, characterizing the corrections by inertial terms to the simple dynamics of Eq. (1). Analogously, Fig. B2 displays the average distances $d_h^W(t)$ and $d_v^W(t)$ as a function of time, characterizing the estimated corrections arising from small scales unresolved by the NEMO velocity field. The effect induced by the inertial terms is very small and clearly negligible. The impact of $\mathbf{W}$, and thus of small unresolved scales is

larger.

    To evaluate the different effects more quantitatively, we summarize in Table B1, considering particles separately in different density ranges (given by the ranges in $\beta$ and associated $v_s$), the average horizontal and vertical pairwise particle distances, calculated after integrating the different dynamics for 10 $days$. To fully appreciate the importance of these numbers, in the two final columns we compute the horizontal and vertical average of the total distance $r$ traveled by the particles (using Eq.

(1)) during the same interval of time. While the influence of the inertial terms is completely irrelevant both in a relative and an absolute sense for any realistic application, more care has to be taken with regards to the unresolved scales. Although the vertical error associated with the latter remains small, its relative importance hugely increases with decreasing settling velocity.

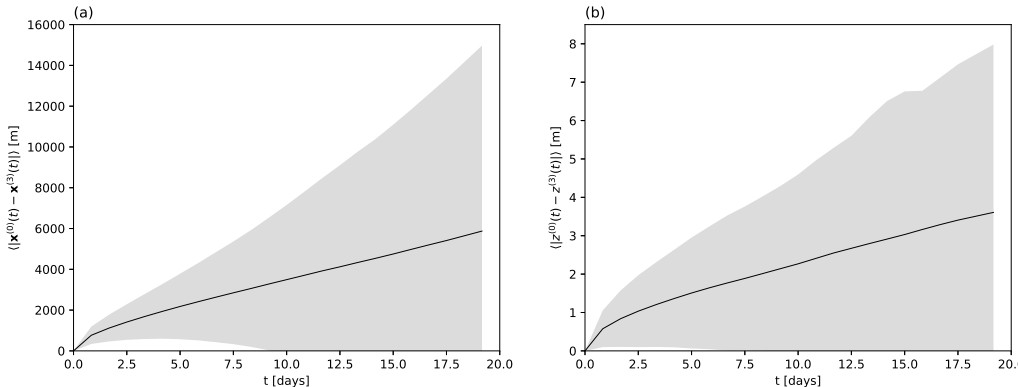

**Figure B2.** Same as Fig. B1 for the comparison of equations (1) and (B2).

**Table B1.** Average horizontal and vertical pairwise particle distances $d$ and single-particle displacements $r$ after an integration time of 10 days. See text. $d/r$ is indicated by percentages in parentheses.

| $\beta$ | $v_s$ ($10^{-3}$ m/s) | $d_h^I$ (m) | | $d_v^I$ (m) | | $d_h^W$ (m) | | $d_v^W$ (m) | | $r_h$ (m) | $r_v$ (m) |
|---|---|---|---|---|---|---|---|---|---|---|---|
| $[0.8, 0.85)$ | $[1.78, 1.25)$ | 60 | (0.26%) | 0.016 | (0.001%) | 2675 | (11.8%) | 2.1 | (0.16%) | 22601 | 1291 |
| $[0.85, 0.9)$ | $[1.25, 0.79)$ | 42 | (0.14%) | 0.010 | (0.001%) | 2831 | (9.7%) | 2.0 | (0.24%) | 29278 | 870 |
| $[0.9, 0.95)$ | $[0.79, 0.37)$ | 31 | (0.08%) | 0.009 | (0.002%) | 3313 | (8.0%) | 2.1 | (0.43%) | 41587 | 493 |
| $[0.95, 1)$ | $[0.37, 0)$ | 16 | (0.03%) | 0.008 | (0.005%) | 4668 | (8.0%) | 2.7 | (1.79%) | 58320 | 151 |

(Indeed, it would tend to infinity for approaching neutral buoyancy.) At the same time, the relative horizontal error is the biggest for the fastest-sinking particles and is well above 10% for them.

We can also observe the time evolution of the $d$ entries of this table in Figs. B3 and B4. The overall effect of the unresolved scales is confirmed to be much larger than that of the inertial terms, and the differences between particles with different densities (ranges of $\beta$) are less noticeable (except for the smallest densities considered, i.e. $\beta \approx 1$).

## Appendix C: The effect of bathymetry on vertical dispersion

We investigate here if the finite and spatially varying depth of the basin (the bathymetry) could affect the conclusions in Sect. 4.4 about vertical dispersion. The falling particles released from different locations reach the seafloor at different times, and then computing some statistics over these particles involves an increasingly narrow set of particles. Note that the bottom boundary (i.e., the seafloor) extends from the surface (close to the coast) to the deepest point of the basin, it is thus relevant at any time during the simulation. We intend to exclude three different effects arising from the continual removal of particles:

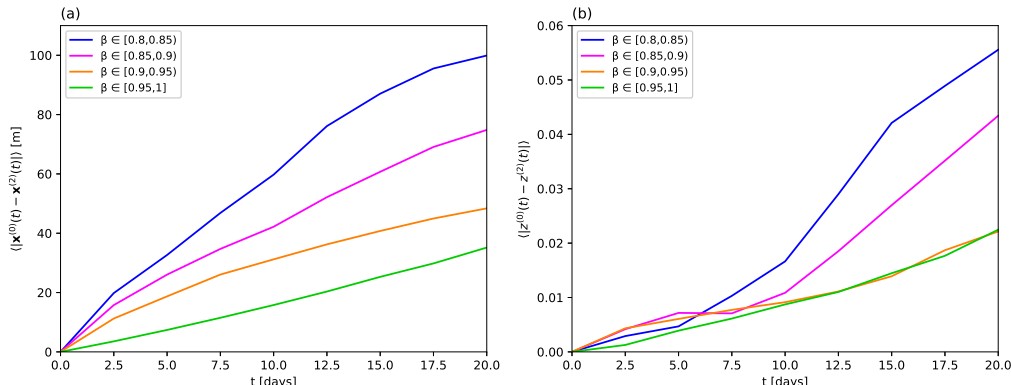

**Figure B3.** Average horizontal distance $d_h^I$ (a) and average vertical distance $d_v^I$ (b) of particles released from the same initial location but integrated with equations (1) and (B1) in the NEMO velocity field. The different lines are obtained from particles from different ranges of densities characterized by the indicated ranges in $\beta$.

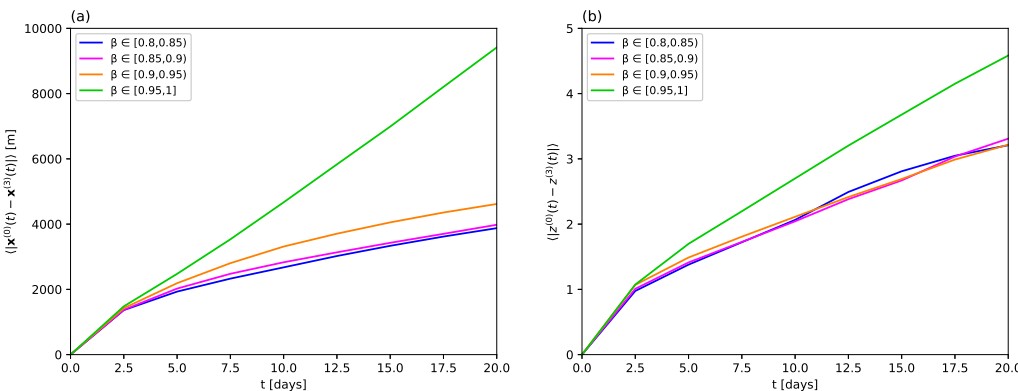

**Figure B4.** Same as Fig. B3 for the comparison of equations (1) and (B2).

(i) distortion of the shape of the distribution close to the boundary, (ii) poor quality of the statistics when many particles have already been lost, (iii) decrease in the geographical area sampled by the particles.

We start with effect (i) by comparing, in Fig. C1, the variance presented in the main text (Fig. 7), computed over all sinking (but not yet sedimented) particles, and that computed over a restricted set of particles. This restricted set contains only those particles at the positions of which the bathymetry $Z$ satisfies $Z > \langle z_i \rangle - 3\sigma_z$, where the average $\langle z_i \rangle$ and the standard deviation $\sigma_z$ are taken with respect to the original (unrestricted) set of particles. This restriction is adaptive and ensures that the seafloor is sufficiently far for its effect on the particle distribution to be negligible at the positions of all particles kept for the computation. Fig. C1 shows that the difference in the results between the full set and the restricted one is negligible for all three settling

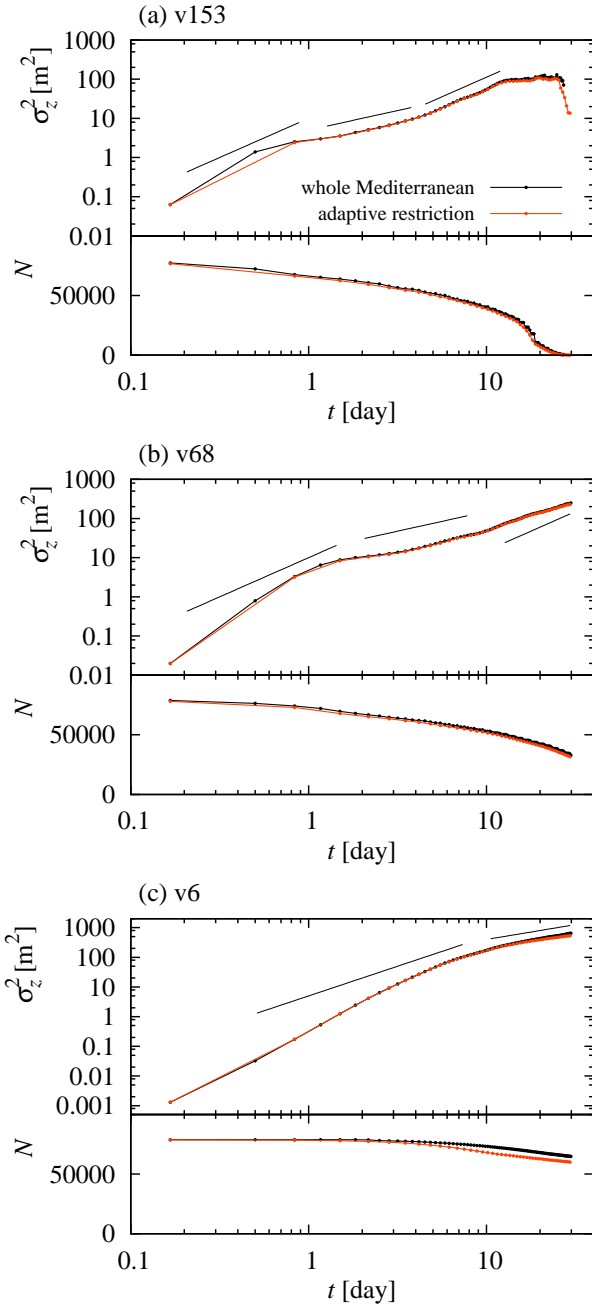

**Figure C1.** Variance of depth reached by all sinking particles (in black) and an adaptively restricted subset of them (in yellow) as a function of time. See text for details. Straight lines represent power laws for reference, with exponents 1 and 2. The number $N$ of particles considered for the computation of the variance is also shown.

velocities considered, except perhaps for the drop at the very end of the time evolution, after the constant section, in Fig. C1a. This drop is, however, very short compared to the bulk of the sinking process and thus have minor importance. Furthermore, the distribution of particles so close to the bottom (cf. Fig. 9) should anyway be strongly influenced by resuspension and remixing by bottom currents (Kane et al., 2020). For the rest of the time evolution, we can be confident that a possible distortion of the particle distribution induced by the boundaries has no impact on the variance curves. Such distortion may be present, but the number of particles close to the seafloor is very small at any given time instant (see the small difference between the numbers $N$ of particles considered for the two kinds of computation in Fig. C1) so that they have a negligible contribution on the originally computed variance. This is a result of the relatively weak dispersion; artifacts might be found for broader distributions, possibly including a hypothetical continuation of the v6 simulation. The time evolution of $N$ in Fig. C1 also assures us that poor-quality statistics, effect (ii), does not arise until the final drop discussed in the previous paragraph. Except for that very final section, the variance is estimated from a sufficiently large number of particles to keep the relative error of the sample variance (with respect to the population variance) very low. This is so because, under standard assumptions, the ratio between the sample and the population variance should be close to a chi-square random variable with $N - 1$ degrees of freedom, divided by $N - 1$ (Douillet, 2009).

The time evolution of $N$ also suggests that effect (iii) is avoided as well: in Fig. C1, there is no sudden drop in the number of particles during the simulations that could result in the changes in the slope of the curves of variance versus time. To further support this conclusion, we compute the time evolution of the variance over the particles initialized in a subregion of the whole Mediterranean, see in Fig. 8. In particular, we choose the box of longitudes $[5, 6.9]$ degrees, and latitudes $[37.5, 42]$ degrees, corresponding to the Sea of Sardinia, where the bathymetry is deep enough to prevent particles from reaching the seafloor (except for the very last few time steps of the v153 configuration), so that the horizontal area sampled by the particles approximately remains constant (remember that horizontal displacements are small compared to geographical features).

According to Fig. C2, the character of the dispersion in the Sea of Sardinia is nearly identical to that in the whole Mediterranean, except after the crossover from normal diffusive to ballistic dispersion in the v153 case (Fig. C2a). The smaller variance in the velocity should naturally lead to a later crossover to the long-time behavior in the Sea of Sardinia (see Section 4.4), it is nevertheless doubtful that the crossover should fall outside the simulation time. As the crossover is not observed, one might speculate that horizontal mixing might just get strong enough to suppress ballistic dispersion, similarly to the v68 case (see the discussion of Fig. 7), for which the time evolution of the variance is very similar in the Sea of Sardinia and the whole Mediterranean (Fig. C2b). Fig. 8 suggests that the characteristic patch size is smaller in the western basin of the Mediterranean, including the Sea of Sardinia, than in the (considerably larger) eastern one, which makes inter-patch mixing easier. We conclude that the only substantial difference between the dispersion in the Sea of Sardinia and the whole Mediterranean may originate from the smaller extension and some special characteristics of the former, and the decrease in the area sampled in the whole-Mediterranean simulation presumably has not effect on the results.

Based on these analyses, we believe that the findings of Section 4.4 are unaffected by the boundary and thus have general relevance for oceanic dispersion.

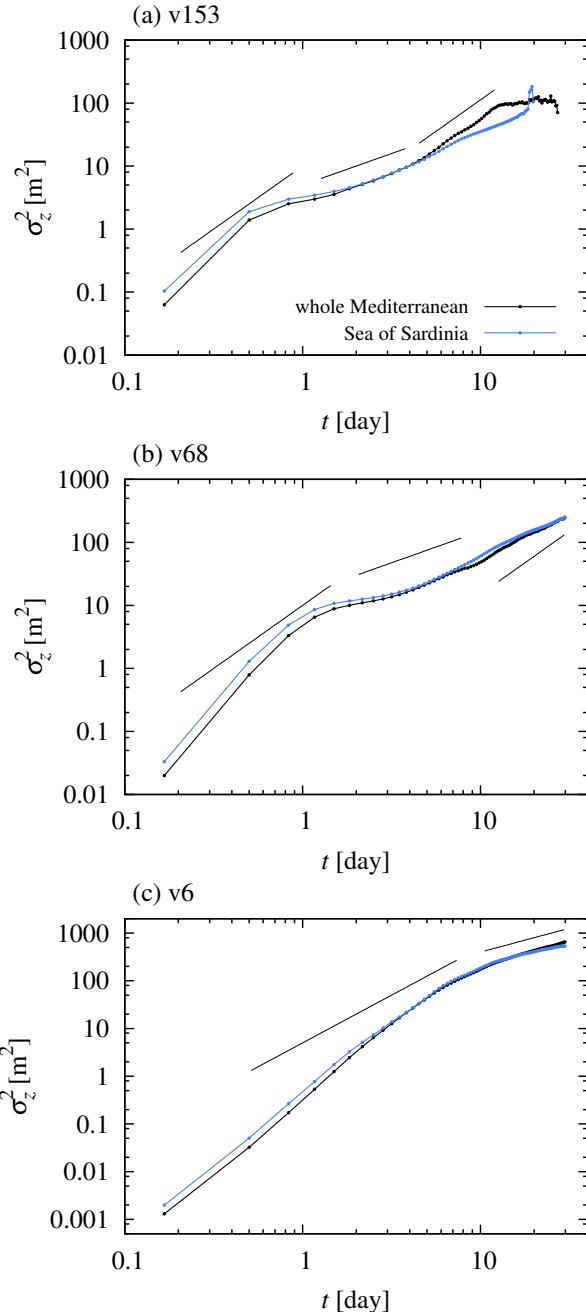

**Figure C2.** Variance of depth reached by all sinking particles (in black) and those initialized in the Sea of Sardinia (in blue; longitudes in [5, 6.9] degrees, and latitudes in [37.5, 42] degrees) as a function of time. See text for details. Straight lines represent power laws for reference, with exponents 1 and 2.

*Author contributions.* All authors designed research. RF performed the simulations. RF and GD analyzed data. RF, GD, EHG and CL prepared the first draft, and all authors reviewed and edited the manuscript.

*Competing interests.* The authors declare that they have no conflict of interest.

*Acknowledgements.* This work is part of the "Tracking of Plastics in Our Seas" (TOPIOS) project, funded by the European Research Council under the European Union's Horizon 2020 research and innovation program (grant agreement no. 715386). We acknowledge publication fee support from the CSIC Open Access Publication Support Initiative through its Unit of Information Resources for Research (URICI). R.F., G.D., E.H-G and C.L acknowledge financial support from the Spanish State Research Agency through the María de Maeztu Program for Units of Excellence in R&D (MDM-2017-0711). R.F. also acknowledges the fellowship no. BES-2016-078416 under the FPI program of MINECO, Spain. G.D. also acknowledges financial support from the European Social Fund through the fellowship no. PD/020/2018 under the postdoctoral program of CAIB, Spain, and from NKFIH, Hungary (grant agreement no. NKFI-124256).

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
