# Peer review of "Sinking microplastics in the water column: simulations in the Mediterranean Sea"

_Ocean Science, 2020_

## Referee Comment (RC1) · Anonymous Referee #1 · 7 Dec 2020

**Reviewer report on ``Sinking microplastics in the water column: simulations in the Mediterranean Sea'' by Fuente et al. (2020)**

**General comments**

This manuscript addresses the transport of negatively buoyant microplastic (spherical) particles in the Mediterranean Sea. The authors have systematically studied the effects of seawater density, unsteady particle acceleration, earth's Coriolis force, and subgrid scale ambient fluid motions on the horizontal and vertical dispersion of particles. A major conclusion is that the particle velocity can simply be taken as the vector sum of the fluid velocity plus a constant settling velocity when basin scale numerical simulations are performed. The authors have also shown that their results are not basin-dependent and should be applicable to other parts of the world.

I found the manuscript to be well-researched and well-versed. The topic is highly relevant to sustainability and ocean cleanup efforts and therefore the manuscript fits well into the scope of Ocean Science. I did not find any major weaknesses or flaws in the methodology and conclusions drawn in the manuscript. The systematic exploration of the roles of various parameters in affecting particle dynamics on the basin-scale is logical and clear.  My recommendation to the editor is accept with minor revision. Below, I provide some suggestions that the authors may consider during their revision.

**Specific comments**

1. I understand the reasons why spherical particles are chosen for analysis in this study. However, rod-like, or other elongated shapes are common in practice. I wonder how the authors' conclusions will change with these particles. I am not looking for a complete analysis but an order-of-magnitude argument. Usually, the Stokes relaxation time is modified to account for the different (as compared to spheres) settling velocity of non-spherical particles. This could be a starting point for the authors to consider.
2. The yellow data lines appearing in figures A3, A4 and B1 should be changed to a different color for better presentation.

---

## Referee Comment (RC2) · Anonymous Referee #2 · 9 Dec 2020

Comments on the manuscript "Sinking microplastics in the water column: simulations in the Mediterranean Sea" by de la Fuente et al.

The authors present presents an important analyzing methodology and tools to study the behavior of the microplastics in the ocean. They specifically focus on the distribution of negatively buoyant rigid microplastic particles in the water column with simulations carried out in the Mediterranean Sea. The modeling tools and methodology followed are reasonably laid out and the paper is well-written. The paper is a useful contribution to the literature in this area. I recommend the publication of the paper after minor revisions suggested below.

1) There is a lack of a discussion on the interaction of these specific rigid microplastic particles considered in the analysis and other particles such as marine snow and

[Figure]

detritus matter that are abundant in the Ocean and transported to the seafloor as aggregates. Microplastics can be scavenged by these settling aggregates. Wouldn't this be an important process that would affect the distribution of microplastics in the water column?

2) Authors show that for fixed horizontal and vertical diffusion coefficient values that are chosen to represent small scale turbulent flow structure has a moderate error. The horizontal diffusion coefficient used 7.25 m2/s seem to be rather large for depths below the mixing layer. Could this have contributed to a large error margin seen in horizontal displacement for the cases shown in Appendix A?

3) Fig A3, A4 caption mentions 'Dashed region indicates the range of the distances among the individual pairs of particles.' I do not see the dashed region referred in these figures.

4) Lines 333-334: 'Both functions have been normalised to have area one, so that they can be displayed in the same plot'. What is area one?

---

## Author Comment (AC1) · 7 Jan 2021

**Supplement to the response letter to RC1**
**Sinking microplastics in the water column: simulations in the Mediterranean Sea**
**(os-2020-95)**

Rebeca de la Fuente, Gábor Drótos, Emilio Hernández-García, Cristóbal López,
and Erik van Sebille

**Deviations from a spherical particle shape**

We quantitatively assess the impact of deviations from a spherical shape through a correction to the settling velocity $v_\mathrm{s}$. The simplified MRG equation, Eq. (3) of the main text or Eq. (A.1) of Appendix A, is 
[revised manuscript text omitted]

**Replacements for Figs. A3, A4 and B1 of the manuscript**

See the new figures as Figs. 2, 3 and 4 of this supplement, displayed after the reference list.

[Figure]

Figure 2: Replacement for Fig. A3 of the manuscript.

[Figure]

Figure 3: Replacement for Fig. A4 of the manuscript.

[Figure]

Figure 4: Replacement for Fig. B1 of the manuscript.

---

## Author Comment (AC2) · 7 Jan 2021

We thank the Reviewer for the careful and positive evaluation. We respond to the specific comments below.

1)

We have checked the literature for detailed descriptions of samples collected from the interior of the water column with results about negatively buoyant rigid microplastic particles. Since field studies of the interior of the water column are not abundant, most of them describe plastic types of positive or unspecified buoyancy from the upper ocean layer (e.g., Egger et al., 2020; Pabortsava and Lampitt, 2020), and the issue of aggregation is usually not addressed, the only relevant publication has been found to be

[Figure]

Bagaev et al. (2017; note that they do describe rigid particles besides microfibres). According to their Section 2.1, selecting microplastic pieces did not require disassembling them from aggregates, in contrast to the case of floating plastics, described in other studies; instead, separate microplastic pieces and organic aggregates were identified. Note that the particles in the class of our interest sink relatively rapidly, and the particulate organic matter content of the water body is concentrated to the surface and is dilute below a few hundred meters' depth (e.g., Karl et al., 1988; Maciejewska and Pempkowiak, 2014). It might be supposed that this could lead to an absence of much interaction with aggregates of biological origin. We should mention, however, experimental results from Michels et al. (2018) showing that suspended polystyrene beads can aggregate with organic matter within a few days in water samples collected from sea surface. This may lead to increased settling velocities (Long et al., 2015), and further research may be needed to fully clarify the degree of aggregation for negatively buoyant rigid microplastics in the water column.

Summarizing the above, we will comment on the issue in Section 2 by adding the following sentences: "Note that, in contrast to the case of floating plastics (Kooi et al., 2017; Kvale et al., 2020), interaction of sinking plastics with particulate matter of biological origin appears to be moderate according to the absence of a need to disassemble microplastic pieces from biological aggregates during sample processing as described by Bagaev et al. (2017). Note, however, that experimental results by Michels et al. (2018) indicate that aggregation with organic material might occur within a sufficiently short time at surface layers, which would likely lead to increased sinking velocities (cf. Long et al., 2015)."

2)

The Reviewer is right. The Okubo formula (Okubo, 1971), from which we obtained 7.25m^2/s for the effective horizontal diffusion coefficient for our grid, is an empirical fit to surface motions, so that it does not describe effective diffusivity at depth. Instead, effective horizontal diffusivity should be weaker below the thermocline according to a

more accurate estimation. In this sense, our results provide an upper bound for the error associated with unresolved scales of fluid motion. We will point this out in a new sentence in Appendix A of the revised version of the manuscript: "Since the Okubo formula is an empirical fit to surface motions, and effective horizontal diffusivity should be weaker below the thermocline, our results provide an upper bound for the error associated with unresolved scales of fluid motion."

3)

The quoted sentence was kept by mistake: we decided not to show that range, because it would deteriorate the visibility of the plots without adding much value. We will delete the sentence in question.

4)

Both functions have been normalized such that the value of their integrals with respect to z is one; in this way the functions can be displayed in the same plot. We will replace the original formulation in the manuscript by this last sentence.

We hope that our responses appropriately address the issues raised by the Reviewer.

References:

Bagaev et al. (2017). Anthropogenic fibres in the baltic sea water column: Field data, laboratory and numerical testing of their motion. Science of The Total Environment 599-600, 560-571. https://doi.org/10.1016/j.scitotenv.2017.04.185

Egger et al. (2020). First evidence of plastic fallout from the North Pacific Garbage Patch. Scientific Reports 10, 7495. https://doi.org/10.1038/s41598-020-64465-8

Karl et al. (1988). Downward flux of particulate organic matter in the ocean: a particle decomposition paradox. Nature 332, 438-441. https://doi.org/10.1038/332438a0

Kooi et al. (2017). Ups and downs in the ocean: effects of biofouling on vertical transport of microplastics. Environmental Science & Technology 51, 7963-7971.

https://doi.org/10.1021/acs.est.6b04702

Kvale et al. (2020). The global biological microplastic particle sink. Scientific Reports 10, 16670. https://doi.org/10.1038/s41598-020-72898-4

Long et al. (2015). Interactions between microplastics and phytoplankton aggregates: Impact on their respective fates. Marine Chemistry 175, 39-46. https://doi.org/10.1016/j.marchem.2015.04.003

Maciejewska and Pempkowiak (2014). DOC and POC in the water column of the southern Baltic. Part I. Evaluation of factors influencing sources, distribution and concentration dynamics of organic matter. Oceanologia 56, 523-548. https://doi.org/10.5697/oc.56-3.523

Okubo (1971). Oceanic diffusion diagrams. Deep Sea Research and Oceanographic Abstracts 18, 789-802. https://doi.org/10.1016/0011-7471(71)90046-5

Pabortsava and Lampitt (2020). High concentrations of plastic hidden beneath the surface of the Atlantic Ocean. Nature Communications 11, 4073. https://doi.org/10.1038/s41467-020-17932-9

---

## Author Response (AR1)

**Authors' response**
**Sinking microplastics in the water column: simulations in the Mediterranean Sea**
**(os-2020-95)**

Rebeca de la Fuente, Gábor Drótos, Emilio Hernández-García, Cristóbal López, and Erik van Sebille

As prompted, we hereby provide a revised manuscript. In this revised version, we have addressed the issues raised by the referees in RC1 and RC2 in the discussion forum. The Referees have not provided new comments since RC1 and RC2. Please find our responses to RC1 and RC2 as AC1 and AC2, respectively, in the discussion forum.

The changes with respect to the previous manuscript version are highlighted in a marked-up version of the revised manuscript and are also listed below.

**Changes in the content**

- We have included a new appendix (Appendix A) about deviations from sphericity, and updated the corresponding content of the main text (in Sections 2 and 4.1).

- We have added a remark in Section 2 about what we know about aggregation with biogenic material for the class of particles of our interest.

- We have included a remark about a weaker effective horizontal diffusivity below the thermocline in new Appendix B (old Appendix A).

**Technical corrections**

- We have made the description of the content of Fig. 5 in Section 4.3 easier to comprehend.

- We have improved visibility in new Figs. B3, B4 and C1 (old Figs. A3, A4 and B1).

- We have corrected the caption of new Fig. B3 (old Fig. A3).

[revised manuscript text omitted]

---

## Author Response (AR2)

**Authors' response**
**Sinking microplastics in the water column: simulations in the Mediterranean Sea**
**(os-2020-95)**

Rebeca de la Fuente, Gábor Drótos, Emilio Hernández-García, Cristóbal López, and Erik van Sebille

25/01/2021

We thank the Topic Editor for the positive evaluation and the indicated considerations. We take into account the comments and provide a revised manuscript. The changes with respect to the previous manuscript are also highlighted in a marked-up version and are described below.

**Changes in the content**

- We have included in Section 4.3 (lines 323-328) a remark about the rapid overturning that occurs in the Gulf of Lions and its possible effect of increasing sinking rates of particles.

- We have cited three references suggested by one reviewer. These citations are located in lines 48-50 and 154-156.